# Sumoylation regulates FMRP-mediated dendritic spine elimination and maturation

Anouar Khayachi[1], Carole Gwizdek[1], Gwénola Poupon[1], Damien Alcor[2], Magda Chafai[1], Frédéric Cassé[1], Thomas Maurin[1], Marta Prieto[1], Alessandra Folci[1], Fabienne De Graeve[3], Sara Castagnola[1], Romain Gautier[1], Lenka Schorova[1], Céline Loriol[1], Marie Pronot[1], Florence Besse[3], Frédéric Brau[1], Emmanuel Deval[1], Barbara Bardoni [4] & Stéphane Martin [4]

Fragile X syndrome (FXS) is the most frequent inherited cause of intellectual disability and the best-studied monogenic cause of autism. FXS results from the functional absence of the fragile X mental retardation protein (FMRP) leading to abnormal pruning and consequently to synaptic communication defects. Here we show that FMRP is a substrate of the small ubiquitin-like modifier (SUMO) pathway in the brain and identify its active SUMO sites. We unravel the functional consequences of FMRP sumoylation in neurons by combining molecular replacement strategy, biochemical reconstitution assays with advanced live-cell imaging. We first demonstrate that FMRP sumoylation is promoted by activation of metabotropic glutamate receptors. We then show that this increase in sumoylation controls the homomerization of FMRP within dendritic mRNA granules which, in turn, regulates spine elimination and maturation. Altogether, our findings reveal the sumoylation of FMRP as a critical activity-dependent regulatory mechanism of FMRP-mediated neuronal function.

[1] Université Côte d'Azur, CNRS, IPMC, 06560 Valbonne, France. [2] Université Côte d'Azur, INSERM, C3M, 06200 Nice, France. [3] Université Côte d'Azur, CNRS, INSERM, iBV, 06108 Nice, France. [4] Université Côte d'Azur, INSERM, CNRS, IPMC, 06560 Valbonne, France. These authors contributed equally: Anouar Khayachi and Carole Gwizdek.  Correspondence and requests for materials should be addressed to S.M. (email: martin@ipmc.cnrs.fr)

In neurons, messenger RNA (mRNA) targeting to synapses and local synthesis of synaptic proteins are tightly regulated. Indeed, dysregulation of such processes leads to structural synaptic abnormalities and consequently to neurological disorders[1] classified as synaptopathies[2]. Among them, the fragile X syndrome (FXS) is the most frequent form of inherited intellectual disablility and a leading monogenic cause of autism with the prevalence of 1:4000 males and 1:7000 females. FXS results from mutations within the *FMR1* gene causing the loss of function of the RNA-binding protein FMRP. Localization studies revealed that FMRP is highly expressed in the central nervous system. FMRP binds a large subset of mRNAs in the mammalian brain and is a key component of RNA granules. These granules transport mRNA along axons and dendrites and are targeted to the base of active synapses to regulate local translation in an activity-dependent manner[3–5]. Therefore, the transport and the subsequent regulation of local translation are critical processes to brain development as they play essential roles in stabilizing and maturing synapses[3,4]. According the role of FMRP in regulating translation at synapses, the loss of FMRP function in FXS leads to a pathological hyperabundance of long thin immature dendritic protrusions called filopodia[6,7]. These structural defects result from an abnormal post-synaptic maturation and/or a failure in the synapse elimination process[8]. An increased number of immature spines associated with severe changes in synaptic transmission and plasticity as well as in social and cognitive behaviors have also been reported in *Fmr1* knockout (*Fmr1*[−/y]) mouse models for FXS[4,9,10].

The majority of FMRP-containing mRNA granules localizes at the base of dendritic spines[3,4]. Neuronal activation leads to the release of mRNAs from dendritic granules and their local translation at synapses (for a review, see ref. [5]). Importantly, this activity-dependent process requires a tight spatiotemporal regulation involving many protein–protein interactions. Such a regulation is mainly governed by post-translational modifications (PTMs). Previous reports have shown that FMRP function is regulated by phosphorylation, which inhibits translation of its associated mRNAs, whereas dephosphorylation of FMRP promotes their translation[11,12]. Activation of metabotropic glutamate receptor 5 (mGlu5R) induces dephosphorylation of FMRP and its subsequent ubiquitination, which ultimately leads to FMRP degradation via the ubiquitin-proteasome pathway[13,14]. Thus, a deeper comprehension of the activity-dependent molecular mechanisms controlling FMRP is absolutely critical to understanding the functional regulation of FMRP-mediated mRNA transport and local protein synthesis in physiological and pathological conditions, including FXS.

Sumoylation is a PTM involved in many cellular signaling pathways. It consists in the covalent enzymatic conjugation of the small ubiquitin-like modifier (SUMO) protein to specific lysine residues of substrate proteins[15,16]. The sumoylation process requires a dedicated enzymatic pathway[17–19]. SUMO paralogs (~100 amino acids; ~11 kDa) are conjugated to its substrates via the action of the E2-conjugating enzyme Ubc9. Sumoylation is a reversible process due to the activity of specific desumoylation enzymes called Sentrin-proteases (SENPs)[20]. At the molecular level, sumoylation can modulate the dynamics of multi-protein complexes by preventing protein–protein interactions and/or by providing new binding sites for novel interactors[21,22].

Sumoylation regulates a wide range of neurodevelopmental processes[18,19,23]. For instance, our group has demonstrated the spatiotemporal regulation of the SUMO system in the developing rat brain[24] and that sumoylation is regulated by neuronal activity[25] and the activation of mGlu5R[26]. Sumoylation also influences various aspects of the neuronal function including neurotransmitter release[27,28], spinogenesis[29,30], and synaptic communication[31–33].

Here, we report that FMRP is a novel sumoylation substrate in neurons. We demonstrate that FMRP sumoylation is absolutely essential to maintaining the shape of mRNA granules in dendrites and to controlling both the spine density and maturation. We identify the active SUMO sites on FMRP and show that activation of mGlu5R rapidly induces FMRP sumoylation triggering the dissociation of FMRP from dendritic RNA granules to allow for local translation. Altogether, our findings shed light on sumoylation as an essential activity-dependent mechanism that tunes spine elimination and maturation in the mammalian brain.

## Results

**FMRP is sumoylated in vivo.** Given the critical importance of FMRP in brain development and maturation, it is of particular interest to understand the molecular mechanisms regulating FMRP function. Thus, we investigated whether FMRP is subjected to sumoylation. To this end, we performed immunoblot analyses and control assays using several commercial as well as in-house anti-FMRP and anti-SUMO1 antibodies on rodent brain homogenates (Fig. 1; Supplementary Fig. 1). We first analyzed rat brain homogenates in absence or presence of NEM (N-ethyl maleimide), which protects proteins from desumoylation during the lysis process[31] (Fig. 1a; Supplementary Fig. 1f). FMRP is detected as isoforms ranging from 70 to 90 kDa. Interestingly, we found a higher molecular weight band at ~120 kDa that was detected only in the presence of NEM (Fig. 1a, total lane). The densitometric analysis of the ratio between the sumoylated form of FMRP and the total level of FMRP in NEM-treated input lanes revealed that there is about 4% of sumoylated FMRP in all the conditions tested (Supplementary Fig. 1c). We confirmed the upper band to be the sumoylated form of FMRP by immunoprecipitation experiments with specific anti-FMRP antibodies and anti-SUMO1 immunoblot (Fig. 1b) or with the converse experiment using anti-SUMO1 immunoprecipitation and anti-FMRP immunoblot (Fig. 1c). We also validated the sumoylation of FMRP in wild-type (WT) mouse brain homogenates (Fig. 1d). Accordingly, we were also able to co-immunoprecipitate the sole SUMO-conjugating enzyme Ubc9 from mouse brain homogenates using anti-FMRP antibodies (Fig. 1e). We further validated the sumoylation of FMRP in vivo using several combinations of FMRP/SUMO1 antibodies (Supplementary Fig. 1d, e, g–j) or in cultured neurons (Supplementary Fig. 1k–n). Immunolabeling experiments (Fig. 1f) showed that FMRP partially co-localizes with Ubc9 and SUMO1 in dendrites of mouse hippocampal neurons, providing further evidence of the interplay between FMRP and the SUMO pathway.

Sumoylation consists in the covalent binding of the SUMO moiety to a lysine residue of the consensus sequence on the substrate protein ($\Psi$KxD/E, where $\Psi$ is a large hydrophobic residue, K is the target lysine, x can be any residue, and D/E are aspartate or glutamate[34]). To identify lysine residues in FMRP potentially targeted by the sumoylation system, we used SUMO-prediction softwares to analyze the primary sequence of FMRP and then alignment tools to assess whether these potential sites are evolutionary conserved across species (Fig. 1g). We identified three conserved residues, two proximal (K88, K130) and one distal (K614) lysines as putative targets of the SUMO system. To validate whether these lysine residues could be sumoylated, we performed site-directed mutagenesis combined with bacterial sumoylation assays[31,35] (Fig. 1h, i). We demonstrated that FMRP sumoylation occurs at these residues (K88, K130, and K614) and showed that their mutation into arginine residues (K-to-R mutation) abolishes the sumoylation of FMRP (Fig. 1h, i). We

confirmed these data using sumoylation assays in mammalian COS7 cells in which the expression of the FMRP-K88,130,614R mutant prevents the sumoylation of FMRP (Fig. 1j). Consistent with the sumoylation of FMRP in the brain and according to our FMRP-SUMO1 models, which were computed using crystal structures available for the N-terminal part of FMRP, both lysine

residues (K88 and K130) are clearly exposed and accessible to sumoylation with a solvent accessible surface area (ASA) of ~70% and ~45%, respectively (Fig. 1k, l). We therefore conclude that FMRP is a SUMO substrate in vivo and that its sumoylation can occur at its N-terminal K88 and K130 and C-terminal K614 residues.

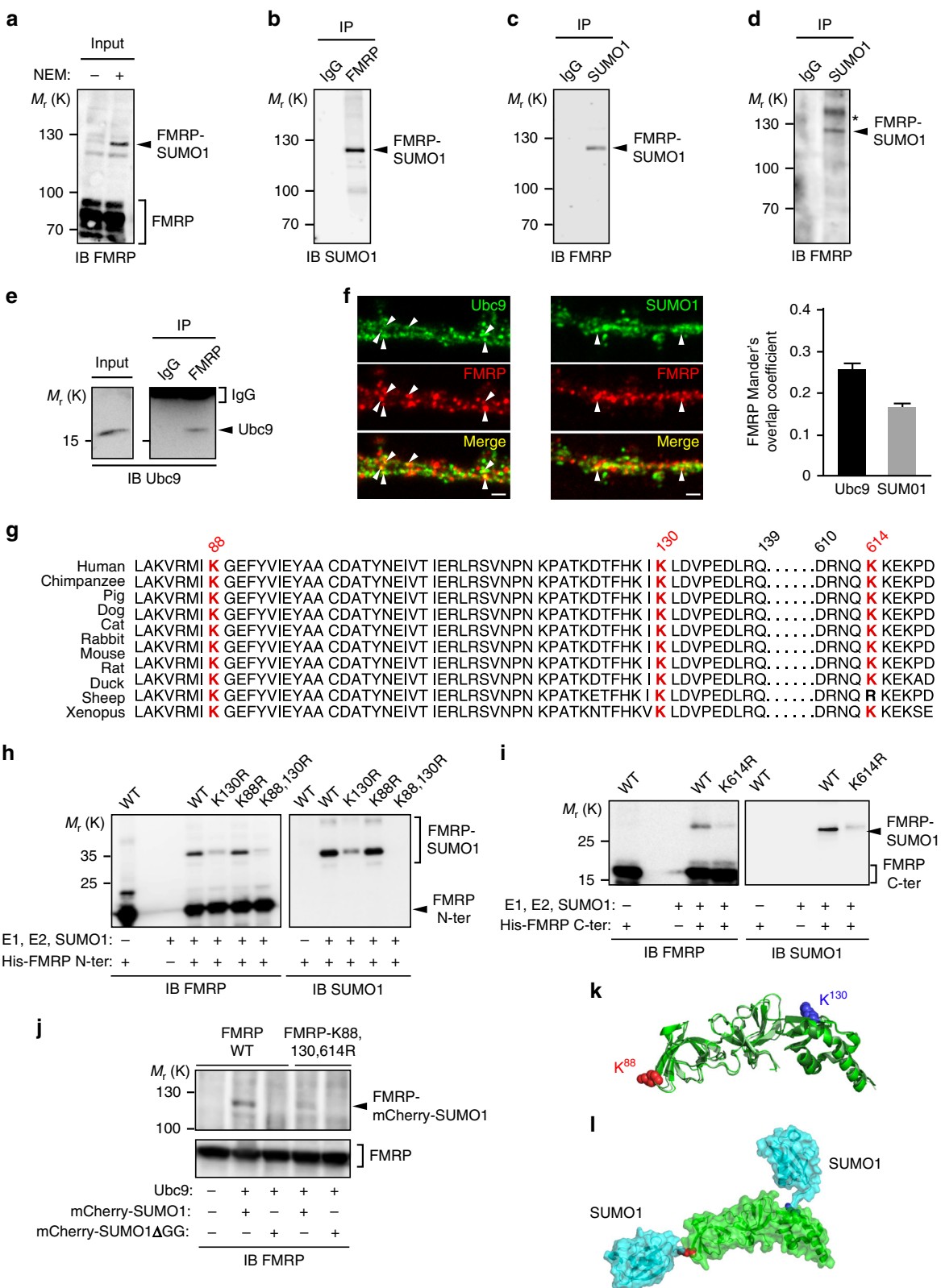

**FMRP sumoylation participates in dendritic spine regulation**. FMRP is essential to proper spine stabilization and maturation[3,4]. In FXS patients, the lack of functional FMRP leads to an immature neuronal morphology with a characteristic excess of abnormally long and thin filopodia[36]. Similar morphological defects are also present in $Fmr1^{-/y}$ mouse brains[37]. Thus, we hypothesized that FMRP sumoylation could be critical in maintaining the density and the maturation of dendritic spines. To address this point, we used attenuated Sindbis particles[38–40] to express either free green fluorescent protein (GFP), the WT GFP-FMRP, the N-terminal K88,130R, C-terminal K614R or non-sumoylatable K88,130,614R GFP-FMRP mutants in cultured $Fmr1^{-/y}$ neurons at 17 days in vitro (17 DIV). We then analyzed and compared the morphology of dendritic spines 24 h post transduction (Fig. 2a, b). In GFP-expressing $Fmr1^{-/y}$ neurons, ~60% of protrusions showed an immature phenotype (see Methods for the spine characterization; Fig. 2a, b). Conversely, the expression of either FMRP WT or the K614R GFP-FMRP mutant, which behaves as the WT, promoted spine maturation (Fig. 2a, b). In stark contrast, expressing either the N-terminal K88,130R or the non-sumoylatable K88,130,614R GFP-FMRP mutant failed to promote spine maturation (Fig. 2a, b).

The excess of dendritic protrusions in neurons is a hallmark of FXS[6,7]. Interestingly, the density of the protrusions was considerably decreased upon the expression of the WT or K614R mutant form of GFP-FMRP (Fig. 2c; GFP control, 7.22 ± 0.16 protrusions per 10 µm; GFP-FMRP WT, 5.34 ± 0.13 protrusions per 10 µm; GFP-FMRP-K614R, 5.39 ± 0.13 protrusions per 10 µm), whereas expressing either the N-terminal K130R, the K88,130R GFP-FMRP mutants, or the non-sumoylatable GFP-FMRP-K88,130,614R did not affect the spine density with measured values almost identical to control neurons expressing free GFP (Fig. 2c). Furthermore, re-expressing WT GFP-FMRP in $Fmr1^{-/y}$ neurons not only affected the spine number but also drastically reduced the mean length of immature spines from ~3.7 µm to <2.6 µm (Fig. 2d).

To individually assess the role of the N-terminal lysine residue, we quantified the morphological changes occuring in $Fmr1^{-/y}$ neurons expressing GFP-FMRP with a single mutated lysine residue (K88R or K130R; Supplementary Fig. 2). While the expression of both mutants promoted spine maturation similarly to GFP-FMRP WT (Supplementary Fig. 2b, d), the K130R mutant failed to reduce the density of the protrusions (Supplementary Fig. 2c; GFP control, 7.22 ± 0.16 protrusions per 10 µm; WT, 5.34 ± 0.13 protrusions per 10 µm; K130R, 6.48 ± 0.15 protrusions per 10 µm) indicating that the integrity of the K130 residue is essential to maintain spine density. Altogether, the data above

indicate that the integrity of both N-terminal lysine residues is critical for the regulation of spine density and maturation since the expression of the K-to-R mutant forms failed to restore the density and the maturity of dendritic spines in $Fmr1^{-/y}$ neurons. Our initial findings therefore support the role of the N-terminal sumoylation of FMRP in the regulation of spine elimination and maturation events.

To start assessing the functional effect of FMRP sumoylation, we compared synaptic transmission by measuring spontaneous miniature excitatory post-synaptic currents (mEPSCs) in $Fmr1^{-/y}$ neurons expressing either GFP-FMRP WT or its non-sumoylatable K88,130,614R mutant (Supplementary Fig. 3). The comparison of cumulative distributions indicated that the amplitude of mEPSCs (from 20 to 40 pA) was significantly increased in neurons expressing the mutant form of GFP-FMRP (Supplementary Fig. 3a, b). Moreover, intervals between mEPSC events (between 300 ms and 1 s) were slightly but significantly increased upon expression of GFP-FMRP-K88,130,614R when compared to GFP-FMRP WT indicating that the mEPSC frequency is decreased in mutant-expressing cells (Supplementary Fig. 3a, c). Data comparing mEPSC properties in WT and $Fmr1^{-/y}$ brain slices have been described in the literature with either a decrease, an increase or no changes in their amplitudes or frequencies, depending on the brain area recorded, the age of the animals, and/or the associated genetic background[41–43]. To our knowledge, there are no available data on mEPSCs recorded from FMRP WT-expressing $Fmr1^{-/y}$ cultured hippocampal neurons and the results from Supplementary fig. 3 indicate that restoring the expression of FMRP in $Fmr1^{-/y}$ neurons leads to changes in basal synaptic transmission, occurring most probably via both pre- and post-synaptic modifications. Additional experiments are now needed to precisely define the associated mechanisms and to address the electrophysiological consequences of FMRP sumoylation in synaptic plasticity in vivo.

**Preventing FMRP sumoylation alters the size of mRNA granules**. Since FMRP is an RNA-binding protein, we also examined whether the mutation of the sumoylation sites interferes with the RNA-binding capacity of FMRP by performing cross-linking and immunoprecipitation (CLIP) assays (Fig. 2e, f). FMRP-CLIPed mRNAs from $Fmr1^{-/y}$ neurons expressing either the WT or K88,130,614R forms of GFP-FMRP were analyzed by quantitative PCR to compare the abundance of some known FMRP target mRNAs (Fig. 2e). Our data showed that either forms of GFP-FMRP are able to bind target RNAs to similar extent (Fig. 2f).

**Fig. 1** FMRP is sumoylated in vivo in the rat and mouse brain and the SUMO system targets the conserved residues K88, 130, and 614 of FMRP. **a** Representative immunoblot anti-FMRP (Ab#056) of P7 post-nuclear rat brain extracts prepared or not in the presence of the cysteine protease inhibitor NEM to prevent desumoylation. **b** Immunoblot anti-SUMO1 of NEM-treated P7 post-nuclear rat brain extracts subjected to immunoprecipitation with FMRP (Ab#056) antibody or control IgG. **c** Converse immunoblot with anti-FMRP (Ab#056) antibody of NEM-treated P7 post-nuclear rat brain extracts subjected to immunoprecipitation with SUMO1 antibody or control IgG. **d** Immunoblot anti-SUMO1 of NEM-treated P1 post-nuclear mouse brain extracts subjected to immunoprecipitation with FMRP (Ab#056) antibody or control IgG. *Non-specific band. **e** Immunoblot of post-nuclear mouse brain extracts (input) subjected to immunoprecipitation with FMRP antibody or control IgG and probed with anti-Ubc9 antibody. **f** Co-localization assays performed on cultured mouse neurons (20 DIV) with antibodies directed against Ubc9, FMRP (Ab#4317), SUMO1. Bar, 2 µm. Degree of co-localization (Manders' coefficient) between FMRP and Ubc9 or SUMO1. N = 3 independent primary cultures with 60 dendrites analyzed for each condition. **g** Sequence alignments showing the evolutionary conservation of the potential SUMO-targeted lysine residues (stars) within the consensus sumoylation sites of FMRP. **h, i** Bacterial sumoylation assay. Representative immunoblots of purified fractions of N- and C-terminal WT or mutated parts of His-FMRP in a recombinant bacterial system and probed with anti-FMRP (**h**, Ab#1C3) or (**i**, #17722) and anti-SUMO1 antibodies as indicated. **j** COS7 sumoylation assay. Immunoblots with anti-FMRP (Ab#056) antibody of full-length WT or lysine-mutated FMRP expressed in COS7 cells with mcherry-SUMO1 WT or mutated (ΔGG) to prevent its conjugation. **k** Original X-ray structures fitted of three human N-terminal FMRP (PDB: 4OVA in green, 4QVZ in light green, 4QW2 in dark green) shown in cartoon representation. K88 and K130 are shown in sphere representation in red and blue, respectively. **l** Original model of FMRP (PDB: 4OVA) and SUMO1 (PDB: 4WJQ) structural links in cartoon and surface representation (with transparency), respectively, in green and light blue. Lysine residues 88 and 130 of FMRP are shown in sphere representation in red and blue, respectively

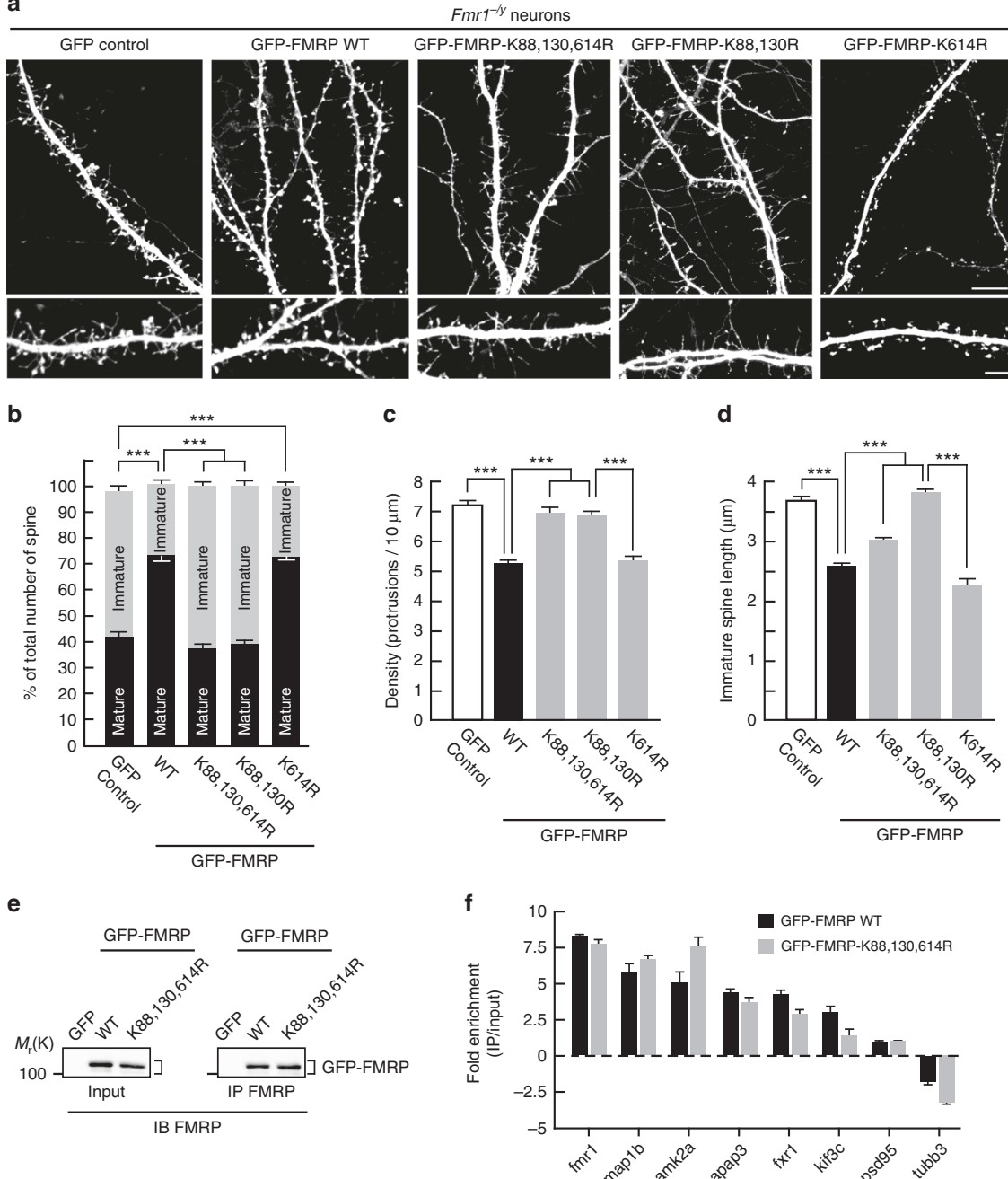

**Fig. 2** The N-terminal sumoylation of FMRP is involved in the regulation of the spine density and maturation. **a** Representative confocal images of dendrites from transduced *Fmr1⁻/y* neurons expressing free GFP, the WT or the non-sumoylatable K88,130,614R, K88,130R, or K614R forms of GFP-FMRP for 24 h. Bar, 10 μm. Enlargements of dendrites are also shown. Bar, 5 μm. Histograms show the relative proportion of mature and immature dendritic spines **b** and the density of the protrusions **c** in GFP, in WT, and mutated GFP-FMRP-expressing cells as shown in **a**. **d** Histograms of immature spine length measured from *Fmr1⁻/y* neurons expressing the indicated constructs. Data shown in **b**–**d** are the mean ± s.e.m. and statistical significance determined by a one-way analysis of variance (ANOVA) with a Bonferonni post-test. $N = \sim 4500$ protrusions per condition from four independent experiments. ***$p < 0.001$. **e, f** CLIP analysis from transduced *Fmr1⁻/y* cortical neurons expressing the WT or the K88,130,614R form of GFP-FMRP revealed that they bind the same RNA repertoire. **e** Representative immunoblots anti-FMRP of the indicated neuronal extracts subjected or not (Input) to immunoprecipitation (IP) with FMRP antibodies. GFP-expressing *Fmr1⁻/y* neurons were used as a negative control. **f** Enrichment (CLIPed/Input) of a set of FMRP-target RNA fragments in the indicated conditions. Several known RNA targets of FMRP (*fmr1*, *map1b*, *camk2a*, *sapap3*, *fxr1*, *kif3c*, and *psd95*) as well as a non-targeted RNA (*tubb3*) were detected by quantitative PCR. Fold enrichment were calculated as described in the Methods section and did not show any statistical differences

Since preventing FMRP sumoylation with the K-to-R mutations does not affect the ability of FMRP to interact with its target RNAs, we hypothesized that FMRP sumoylation is involved in the transport of mRNAs along dendrites. To this purpose, we first examined the FMRP-containing granules along dendrites. We transfected *Fmr1⁻/y* neurons to express either the WT or K88,130R form of GFP-FMRP and performed smFISH experiments using Stellaris probes complementary to three known FMRP mRNA targets: GFP (for GFP-FMRP), PSD-95[44], and CaMKII mRNAs (Fig. 3a–c). Interestingly, the fluorescence of all

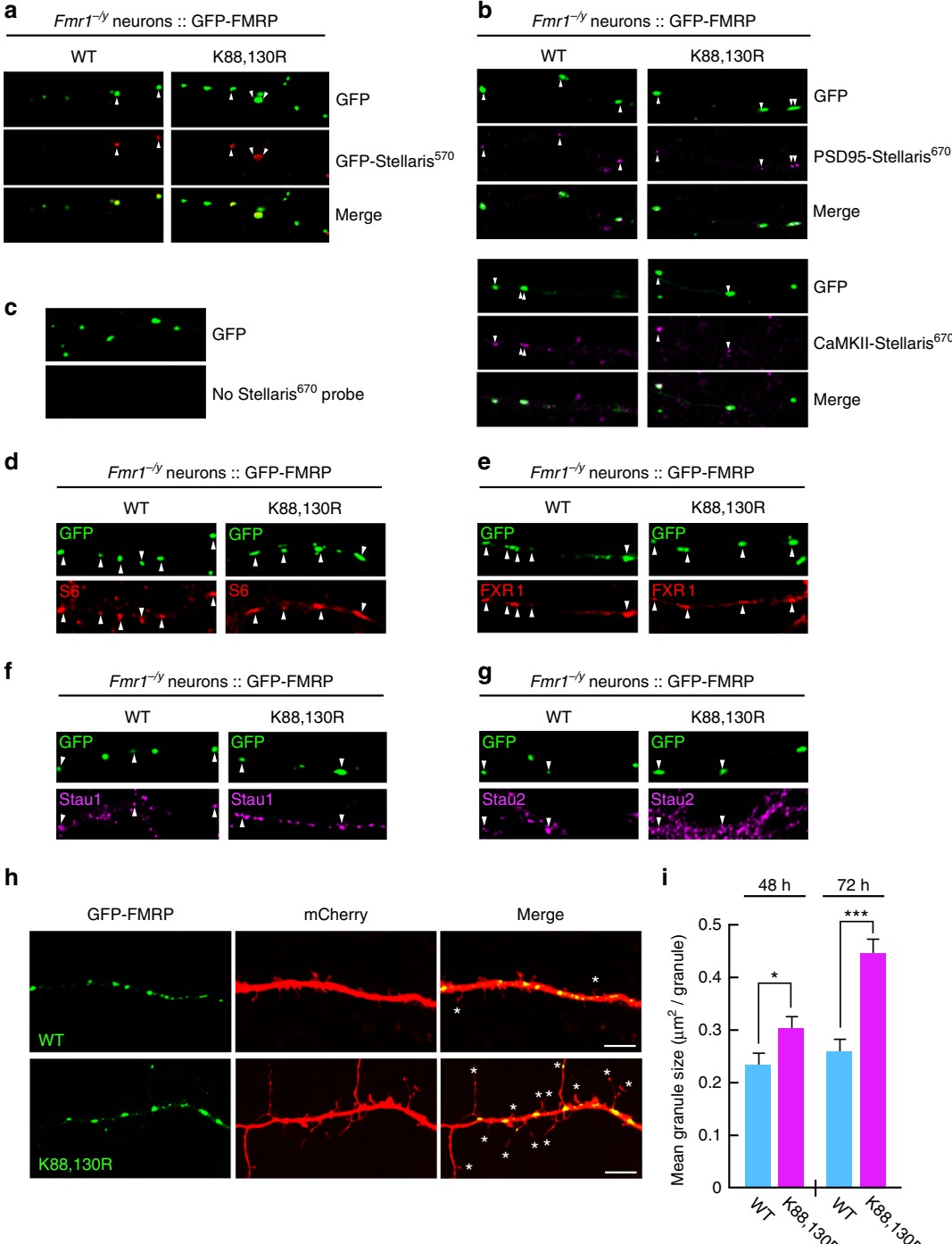

**Fig. 3** Preventing FMRP sumoylation drastically impacts on the size of dendritic FMRP-containing mRNA granules. **a–c** Representative images of WT and K88,130R GFP-FMRP-expressing *Fmr1^−/y^* dendrites were hybridized with GFP **a**, PSD-95 **b**, or CaMKII mRNA **b,** using Stellaris probes. Arrowheads show the co-localization between the indicated Stellaris signals and the GFP-FMRP granules. **c** GFP-FMRP-transfected neurons with no Stellaris probes were used as FISH controls. **d–g** Co-localization assays performed on WT and K88,130R GFP-FMRP-expressing *Fmr1^−/y^* neurons with antibodies directed against the S6 ribosomal protein **d**, FXR1 **e**, and the RNA-binding proteins Staufen 1 **f** and Staufen 2 **g**. Arrowheads indicate the co-localization with the GFP-FMRP positive mRNA granules. **h** Representative confocal images of dendrites from co-transfected *Fmr1^−/y^* neurons co-expressing free mCherry with either the WT or the K88,130R form of GFP-FMRP for 72 h. Bar, 5 μm. **i** Histograms show the mean size of dendritic GFP-FMRP granules after 48 and 72 h of expression. $N = 190–460$ granules per condition from three to four separate experiments. Data shown in **i** are the mean ± s.e.m. and statistical significance was determined using unpaired *t* test. *$p < 0.05$; ***$p < 0.0001$

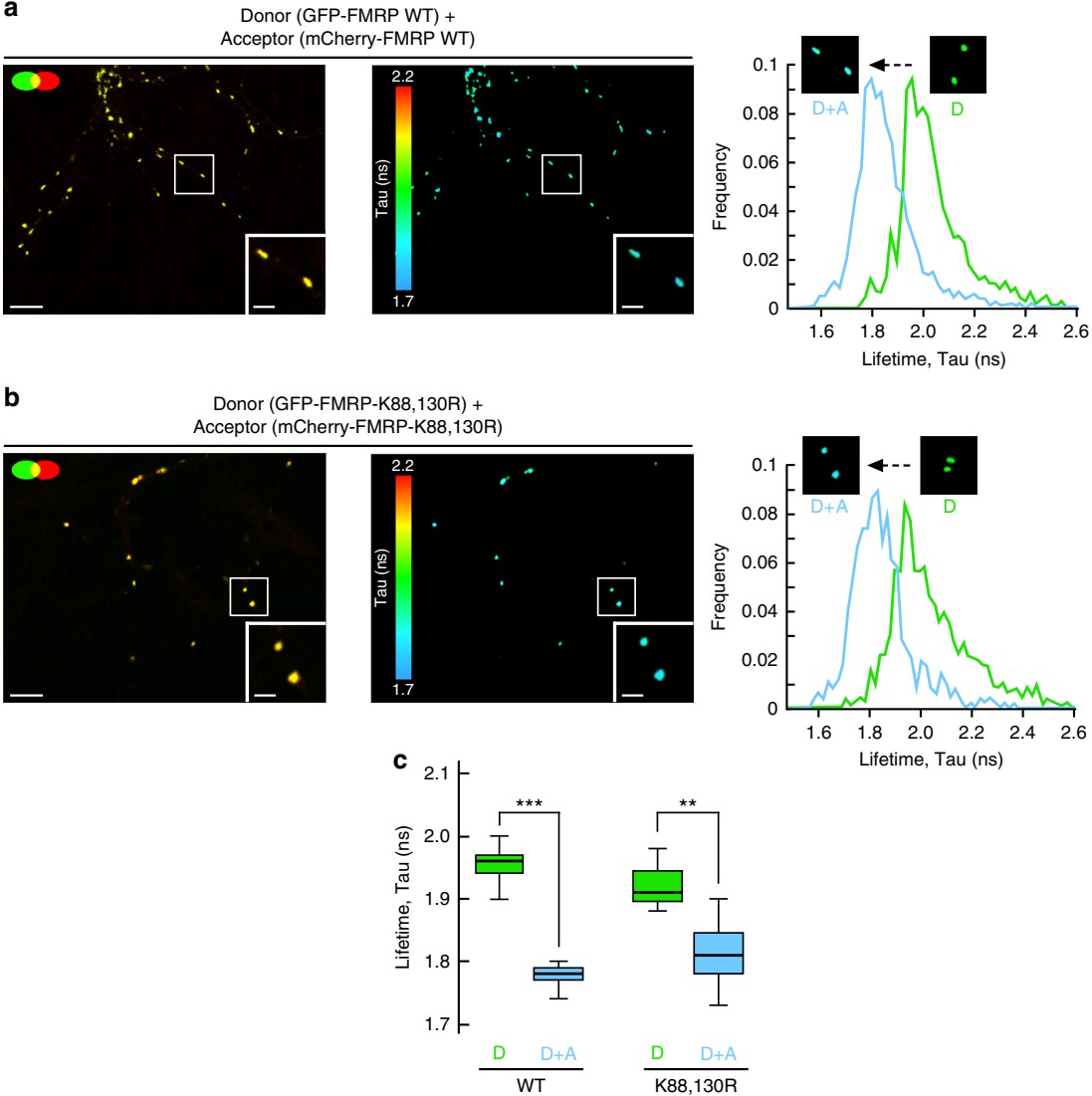

**Fig. 4** Preventing the N-terminal sumoylation of FMRP by the K88,130R mutation does not alter the homomeric FMRP–FMRP interaction within dendritic mRNA granules. **a**, **b** Analysis of GFP-FMRP/mCherry-FMRP interaction within dendritic mRNA granules by fluorescence lifetime imaging (FLIM). Representative confocal images showing the co-localization of the WT **a** or the K88,130R **b** forms of GFP-FMRP and mCherry-FMRP (left images) in dendritic granules; bar, 4 μm. FLIM images of the same field are shown on the right images **a**, **b** where fluorescence lifetime is represented using a pseudo-color scale ranging from 1.7 to 2.2 ns. Insets show representative clusters for each condition; bar, 1 μm. The third row represents the distribution histograms of GFP-FMRP fluorescence lifetime of the donor (D) alone in green and the donor + acceptor (D + A) in blue. FLIM images corresponding to the donor alone condition are displayed in Supplementary Fig. 3b. **c** Box and whiskers plots show the variation of the lifetime determined from FLIM curves. This representation displays upper and lower quartiles, maximum and minimum values in addition to median. $N = 114$–189 granules per condition from three separate experiments. Statistical significance in **c** was determined by a non-parametric Mann–Whitney test. **$p < 0.01$; ***$p < 0.0001$

three probe sets was detectable in GFP-positive granules from secondary dendrites containing either the WT or mutant K88,130R form of GFP-FMRP (Fig. 3a–c). Together with the CLIP experiments (Fig. 2e, f), this reveals that both WT and K88,130R GFP-FMRP-containing granules can travel along dendrites, carrying their mRNA cargoes.

We further characterized these mRNA granules using co-localization assays to investigate whether known components of such granules[45,46] are also present in WT and K88,130R-GFP-FMRP positive granules. As clearly depicted in Fig. 3d–g, both the WT and K88,130R GFP-FMRP granules co-localize with the ribosomal protein S6 (Fig. 3d) and the RNA-binding proteins FXR1 (Fig. 3e), Staufen 1 (Fig. 3f), and Staufen 2 (Fig. 3g), indicating that these granules contain not only some of the target

mRNAs of FMRP (Fig. 3a–c) but also several described components of such dendritic mRNA granules[45,46].

We then measured the surface of dendritic GFP-FMRP-positive mRNA granules at different time points post transfection (Fig. 3h, i). Interestingly, the expression of the K88,130R GFP-FMRP for 48 h significantly increased the size of FMRP-containing granules compared to the WT GFP-FMRP-positive granules (Fig. 3i; WT 48 h, 0.236 ± 0.017 μm$^2$; K88,130R 48 h, 0.305 ± 0.020 μm$^2$). The difference in granule size between the WT and the K88,130R form of GFP-FMRP was further enhanced after 72 h of transfection (Fig. 3i; WT 72 h, 0.265 ± 0.020 μm$^2$; K88,130R 72 h, 0.440 ± 0.030 μm$^2$). All these data reveal that the expression of GFP-FMRP K88,130R results in larger FMRP-containing dendritic mRNA granules suggesting that FMRP

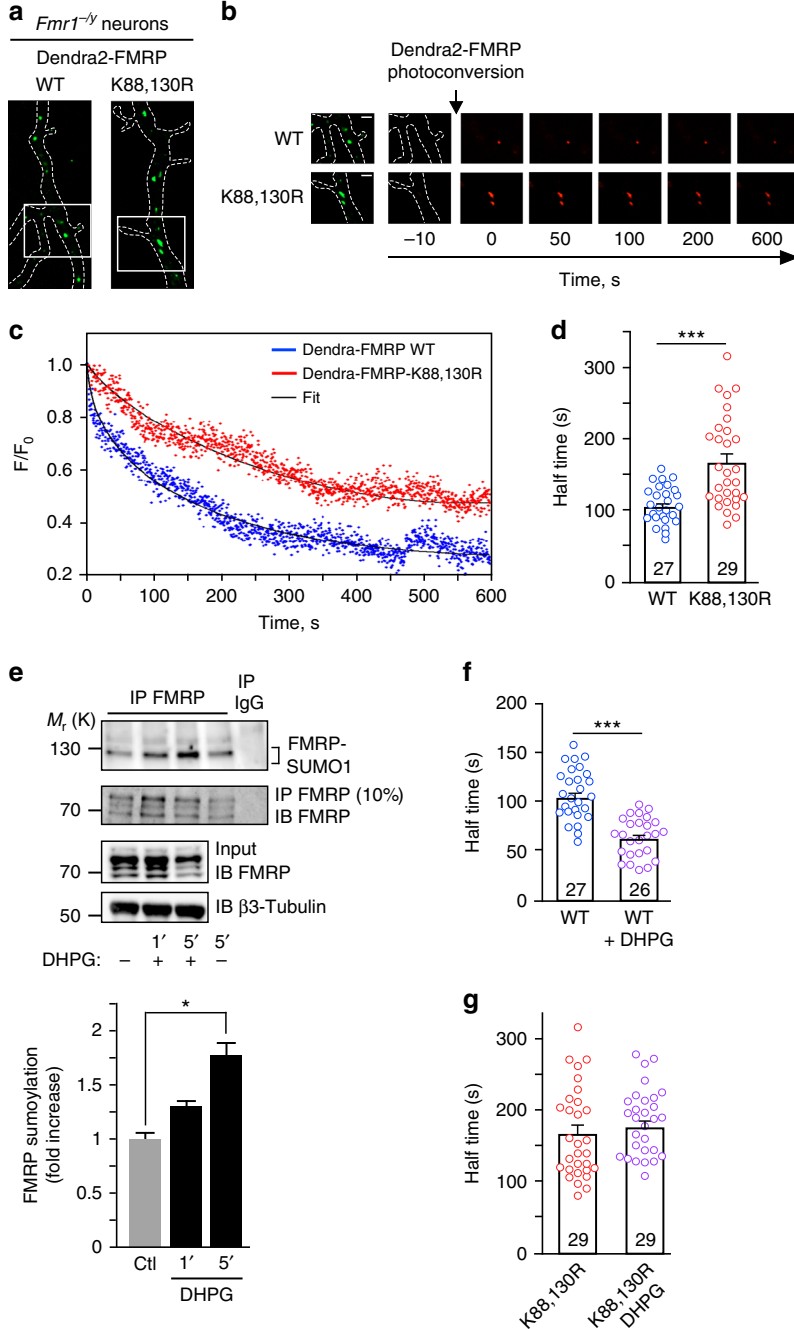

**Fig. 5** Activation of mGlu5 receptors promotes FMRP sumoylation and leads to the release of FMRP from dendritic mRNA granules. **a** Images of transfected $Fmr1^{-/y}$ dendrites expressing the WT or the non-sumoylatable K88,130R forms of GFP-FMRP before Dendra2-FMRP photoconversion are shown. **b** Time lapse series of confocal images of photoconverted Dendra2-FMRP red fluorescence in dendritic granules in basal unstimulated conditions. Enlargement of dendritic granules from the boxed area in **a** is also shown on the left. The decrease in red photoconverted Dendra2-FMRP fluorescence was then monitored over time. Scale bar, 1 μm. **c** Representative sample recording traces of normalized fluorescence from photoconverted WT or mutated Dendra2-FMRP in individual granules in basal unstimulated conditions. The thin traces (black) represent the corresponding fits. **d** Histograms with scatter plots of computed half-time of photoconverted WT and K88,130R Dendra2-FMRP fluorescence diffusion in granules in basal conditions. The number of photoconverted granules is indicated on the bars. **e** Immunoprecipitation of FMRP (Ab#046) and immunoblotting for SUMO1. Control for the immunoprecipitated FMRP fractions is also depicted. Input lanes for FMRP and β3-tubulin are also shown. Quantification for DHPG-induced endogenous FMRP sumoylation in neurons over time is also indicated. The data are from three separate experiments and show the mean ± s.e.m. *$p = 0.0213$. **f** Histograms with scatter plots of half-time of photoconverted Dendra2-FMRP WT fluorescence diffusion in granules from $Fmr1^{-/y}$ neurons stimulated with DHPG. The number of photoconverted granules is indicated on the bars and the histogram/scatter plot in absence of stimulation is taken from **d**. **g** Histograms with scatter plots of half-time of photoconverted Dendra2-FMRP-K88,130R fluorescence diffusion in granules in basal and DHPG-stimulated conditions. The histogram/scatter plot in absence of stimulation is taken from **d**. The number of photoconverted granules is indicated on the bars. Data shown in **d**–**f** and **g** are the mean ± s.e.m. Statistical significance in **d**, **f**, and **g** was determined using a non-parametric Mann–Whitney test. Statistical significance in **e** was determined by an ANOVA with a Bonferroni post-test. *$p < 0.05$; ***$p < 0.0001$

sumoylation could participate in the regulation of FMRP interactions within these granules.

FMRP has been reported to form homodimers via its N-terminal 1–134 domain[47], where the sumoylatable K88 and K130 residues are localized. Thus, to assess whether the difference in granule size measured in Fig. 3i results from abnormal interaction properties of FMRP homodimers directly inside dendritic granules, we performed fluorescence lifetime imaging microscopy (FLIM) experiments on neurons co-expressing WT or K88,130R GFP-FMRP with their respective WT or K88,130R mCherry-tagged constructs (Fig. 4; Supplementary Fig. 4). We observed a clear co-localization of the mCherry/GFP-FMRP constructs in dendritic granules confirming the incorporation of the proteins into granules (Fig. 4a, b). The energy transfer known as fluorescence resonance energy transfer from donor green fluorescent protein (GFP) toward the acceptor mCherry is

quantified by the reduction of the donor fluorescence lifetime (Fig. 4c). We measured a significant reduction of the donor GFP-FMRP fluorescence lifetime in presence of mCherry-FMRP indicating that FMRP/FMRP interaction occurs in dendritic granules. Interestingly, we also found that this homomeric interaction is not affected by the K88,130R mutations (Fig. 4c).

**Sumoylation triggers FMRP dissociation from mRNA granules.** Our results so far indicate that preventing FMRP sumoylation directly impacts on the morphology of mRNA granules in dendrites (Fig. 3h, i) without altering the intrinsic FMRP/FMRP interacting properties within the granules (Fig. 4). Therefore, we investigated whether the absence of FMRP sumoylation affects the dissociation of FMRP from dendritic granules. To assess the diffusion properties of FMRP in dendritic granules, we performed

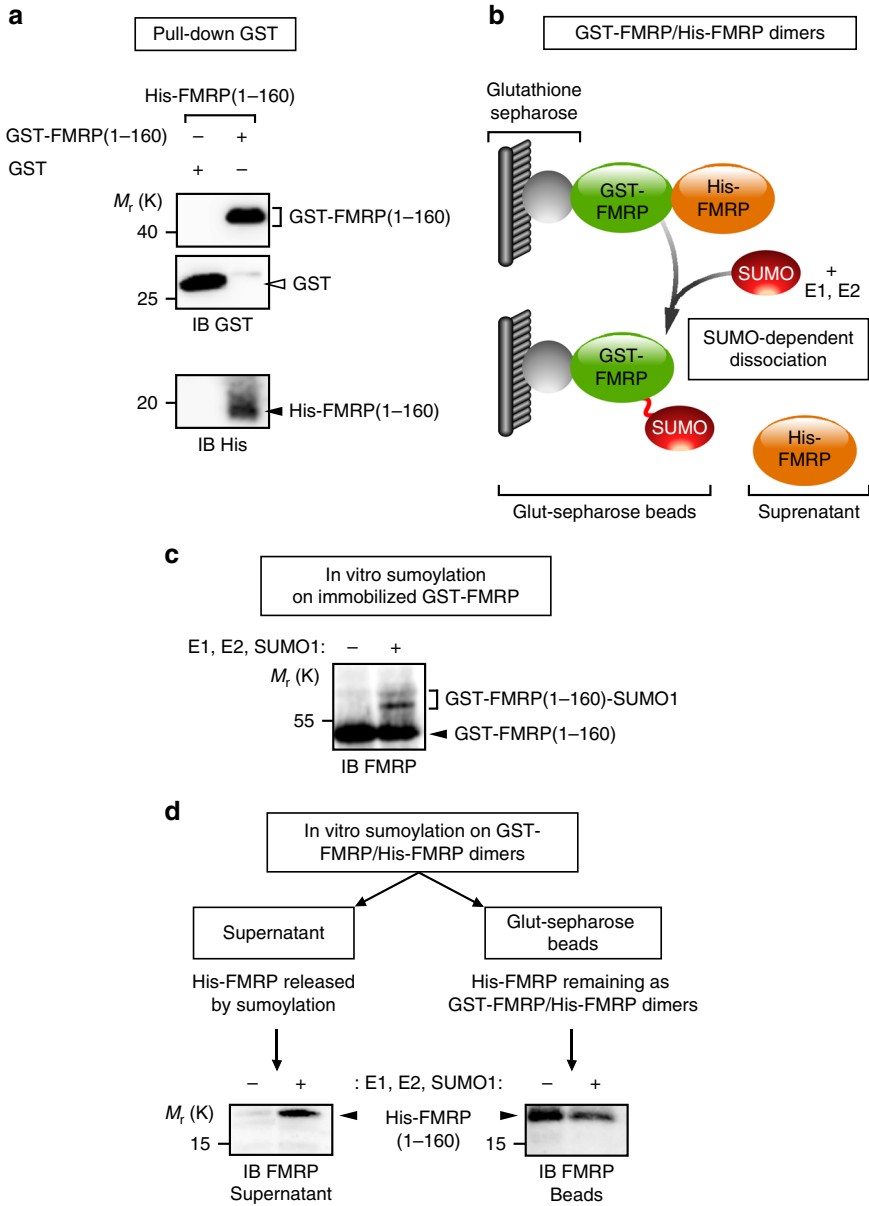

**Fig. 6** The N-terminal sumoylation of FMRP dissociates FMRP homomers. **a** GST pull-down of purified His-FMRP (1–160 aa) with the N-terminal (1–160 amino acids) domain of FMRP fused to the GST protein. Free GST is used as a negative control. **b** Schematic diagram of the SUMO-dependent dissociation assay showing the release into the supernatant of His-FMRP from the immobilized sumoylated GST-FMRP fraction. **c** In vitro sumoylation assay on immobilized GST-FMRP (1-160 aa). **d** In vitro sumoylation assay on GST-FMRP/His-FMRP dimers. Representative immunoblots anti-FMRP (Ab#2F5–1) following the SUMO-dependent dissociation of His-FMRP

live-time restricted photoconversion experiments[48] in *Fmr1*[−/y] neurons expressing the photoswitchable WT or K88,130R Dendra2-FMRP constructs (Fig. 5a, b). Dendra2 is a green-to-red photoactivatable fluorescent protein that allows the real-time tracking of a photoconverted protein[49,50]. We measured and compared the half-times of the decrease in red photoconverted fluorescence, which corresponds to the real-time diffusion of WT and K88,130R Dendra2-FMRP out of dendritic granules (Fig. 5b–d). In basal conditions, the mean half-time of Dendra2-FMRP WT fluorescence dissociation from dendritic granules was significantly shorter than the value measured for the Dendra2-FMRP K88,130R mutant (Fig. 5d; half-time WT = 101.8 ± 4.5 s vs half-time K88,130R = 165.3 ± 12.1 s) indicating that the dissociation of WT FMRP from the granules is much faster than for

the K88,130R mutant. These data strongly support the involvement of FMRP sumoylation in controlling the dissociation of the protein from dendritic mRNA granules.

Activation of mGlu5R regulates FMRP-mediated mRNA transport[51,52] and also modulates its phosphorylation and ubiquitination[13,14]. Interestingly, we previously showed that activation of these receptors also evokes sumoylation in cultured neurons[26]. This prompted us to assess whether the application of the mGluR agonist DHPG triggers FMRP sumoylation in neurons (Fig. 5e). We first confirmed that the activation of mGlu5R with DHPG is effective in our neuronal cultures and evokes an intracellular calcium increase (Supplementary Fig. 5). Then, FMRP-immunoprecipitates were probed with specific anti-SUMO1 antibodies and revealed that the sumoylation of FMRP

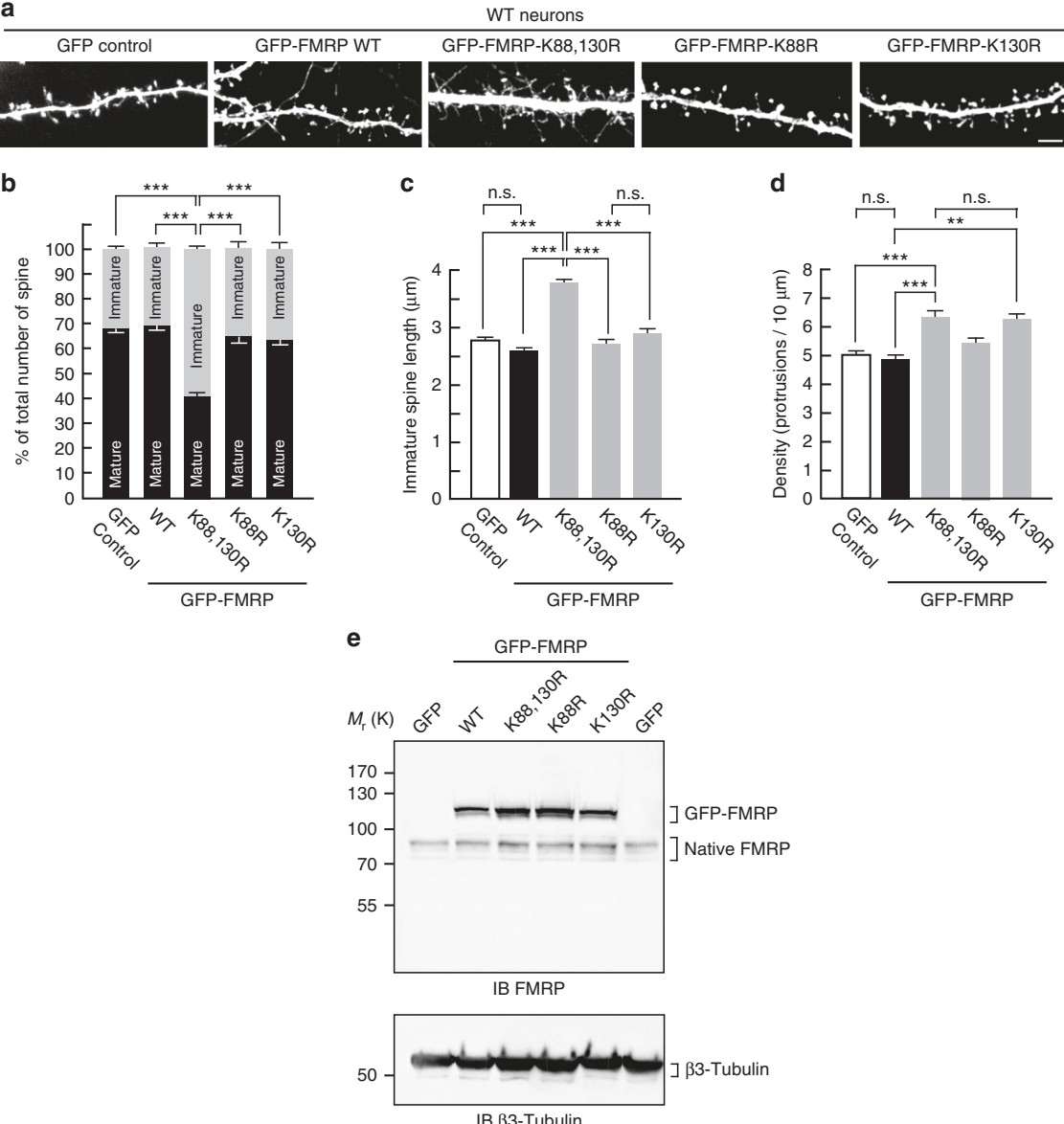

**Fig. 7** Spine density and maturation processes are intrinsically linked to the ability of FMRP to be sumoylated. **a** Representative confocal images of dendrites from transduced WT neurons expressing free GFP, the WT, K88R, K130R, or K88,130R mutant forms of GFP-FMRP for 30 h. Bar, 5 μm. Histograms show the relative proportion of mature and immature spines **b** and the density of the protrusions **d** in the indicated conditions shown in **a**. **c** Histograms of immature spine length measured from WT neurons expressing the indicated constructs. **e** Relative protein expression levels of the WT and mutant forms of GFP-FMRP in WT transduced neurons as in **a** showing an approximate threefold increase in the levels of exogenous GFP-FMRP expression. Data shown in **b**–**d** are the mean ± s.e.m. Statistical significance in **b**–**d** was determined by a one-way analysis of variance (ANOVA) with a Bonferroni post-test. N = ~3000 spines per condition from four independent experiments. ***p < 0.001; n.s. not significant

is low in basal unstimulated conditions but rapidly increases after 1 and 5 min of DHPG treatment (DHPG 1 min, 1.28 ± 0.12 fold/control; DHPG 5 min, 1.73 ± 0.2 fold/control; Fig. 5e) indicating that FMRP sumoylation is rapidly triggered by the mGlu5R activation.

These results led us to hypothesize that the activity-dependent sumoylation of FMRP controls FMRP dissociation from dendritic mRNA granules. To address this point, we pharmacologically stimulated mGlu5R in $Fmr1^{-/y}$ neurons expressing either Dendra2-FMRP WT or K88,130R and measured the dissociation properties of FMRP from dendritic granules using the photo-conversion assay (Fig. 5f, g). Interestingly, mGlu5R stimulation enhanced the exit rate of the red photoconverted Dendra2-FMRP WT fluorescence from granules by ~40% (Fig. 5f). By contrast, mGlu5R activation had no effect on the dissociation of Dendra2-FMRP-K88,130R positive granules (Fig. 5g). These findings strongly support that the mGlu5R-dependent sumoylation of FMRP regulates the dissociation of FMRP from dendritic mRNA granules.

**Sumoylation regulates homomeric FMRP–FMRP interaction.** Our data demonstrate that FMRP sumoylation controls FMRP release from dendritic granules. To further assess the role of sumoylation in the regulation of FMRP–FMRP interaction, we combined pull-down assays with in vitro SUMO reactions and analyzed the impact of sumoylation on the dissociation of FMRP homomers (Fig. 6).

We purified GST- and His-tagged FMRP (1–160 aa) fusion proteins and found that GST-FMRP (1–160) specifically interacts with His-FMRP (1–160 aa) and forms N-terminal FMRP homodimers in vitro (Fig. 6a). We then performed an in vitro sumoylation assay[31] on purified FMRP (1–160 aa) dimers to assess whether sumoylation promotes their dissociation (Fig. 6b–d). First, we verified that the immobilization of GST-FMRP (1–160 aa) on the glutathione matrix did not prevent the in vitro sumoylation of the protein (Fig. 6c). Incubation of immobilized GST-FMRP (1–160 aa) with the sumoylation reaction mix gave rise to higher molecular weight bands corresponding to the sumoylated forms of GST-FMRP (1–160 aa). These bands were absent in control conditions (Fig. 6c).

Next, we performed in vitro sumoylation assays on immobilized GST-FMRP–His-FMRP dimers (Fig. 6d). The pool of His-FMRP (1–160 aa) released by sumoylation was separated from the remaining immobilized dimers by centrifugation of the glutathion beads. Proteins either in the supernatant or bound to the beads were both analyzed by immunoblotting with anti-FMRP antibodies. As seen in Fig. 6d, the release of His-FMRP

(1–160 aa) from the immobilized dimers was only promoted upon sumoylation with the concurrent decrease of the remaining His-FMRP (1–160 aa) in the pelleted FMRP fraction. This particular set of data demonstrates that sumoylation promotes the dissociation of FMRP–FMRP dimers.

**SUMO-deficient FMRP-expressing WT neurons show FXS phenotype.** Collectively, our data clearly demonstrate that sumoylation of the N-terminal part of FMRP is essential to allow for the dissociation of the protein from dendritic mRNA granules and to promote spine elimination and maturation. To confirm the key involvement of FMRP sumoylation in neuronal maturation events, we hypothesized that the expression of the non-sumoylatable FMRP mutant could reverse the spine density and maturation of WT neurons. Thus, we expressed either the WT or the K88,130R mutant form of GFP-FMRP into WT mouse neurons (Fig. 7). WT neurons expressing GFP-FMRP-K88,130R resembled the GFP-expressing $Fmr1^{-/y}$ neurons (Fig. 2) with >67% of protrusions characterized by an immature phenotype (Fig. 7a, b). Similarly, the length of dendritic spines in WT neurons expressing GFP-FMRP-K88,130R was also significantly increased (Fig. 7c; K88,130R, 3.77 ± 0.08 μm) comparable to the values measured in $Fmr1^{-/y}$ neurons (Fig. 2d).

Importantly, the density of dendritic spines was dramatically increased upon the expression of the K88,130R mutant (Fig. 7a, d; GFP control, 5.03 ± 0.17 protrusions per 10 μm; K88,130R, 6.33 ± 0.24 protrusions per 10 μm), comparable to the values obtained in $Fmr1^{-/y}$ neurons (Fig. 2c). Interestingly, the expression of the single K130R mutant in WT mouse neurons also leads to a significant increase in the density of the protrusions (Fig. 7a, d; GFP control, 5.03 ± 0.17 protrusions per 10 μm; K130R, 6.29 ± 0.41 protrusions per 10μm) without altering the maturity of dendritic spines (Fig. 7b, c). As expected, expressing the WT form of GFP-FMRP in WT neurons did not affect any of the spine characteristics confirming the essential role of FMRP sumoylation in spine elimination and maturation processes.

## Discussion

Here, we report for the first time that FMRP is a sumoylation target in vivo. We identify three sumoylatable residues, two of which lay within the N-terminal domain of FMRP and are the active SUMO sites. We further find that the activation of metabotropic mGlu5R promotes the sumoylation of FMRP and rapidly leads to the dissociation of FMRP from dendritic mRNA granules allowing for the regulation of spine elimination and maturation (Fig. 8). Thus, our work uncovers a novel activity-

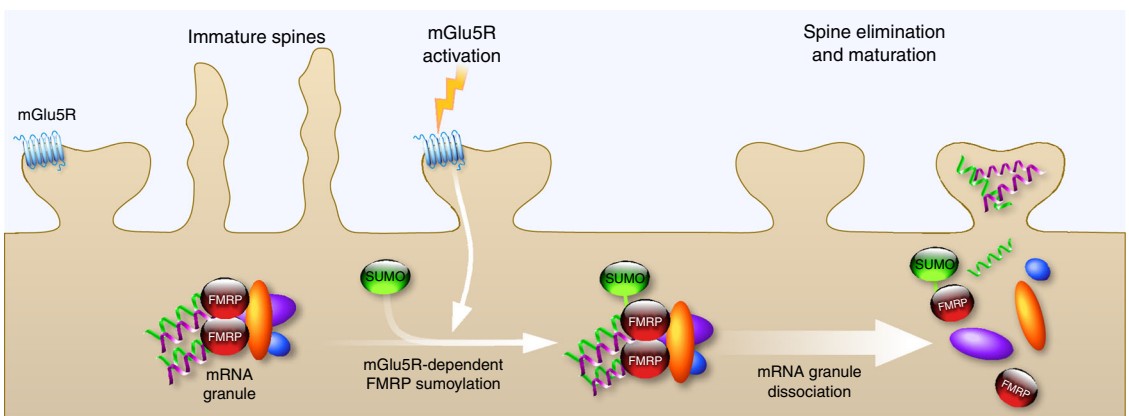

**Fig. 8** Schematic model for the mGlu5R-dependent regulation of FMRP function via the sumoylation process. The activity-dependent sumoylation of FMRP is a key step to dissociate FMRP from dendritic mRNA granules and consequently to regulate spine elimination and maturation

dependent role of sumoylation in the regulation of FMRP neuronal function.

We provide the first evidence that FMRP sumoylation is required for spine elimination and proper maturation. The initial step of spine formation is the emergence of immature long thin protrusions, which are later on eliminated or matured with enlargement of spine head[8]. A tight balance between these processes is thus required for the development of a functional neuronal network. This is in line with our data showing a decrease in the density of protrusions when expressing FMRP in $Fmr1^{-/y}$ neurons, and an increased density in WT neurons expressing the SUMO-deficient form of FMRP. Such compensatory and deleterious effects support the idea that immature spines are overproduced and/or less efficiently eliminated when FMRP sumoylation is perturbed.

In correlation with our findings, the role of sumoylation at the post-synaptic compartment has already been described for several proteins[19]. For instance, sumoylation of the scaffolding calcium/calmodulin-dependent serine protein kinase (CASK) reduces CASK interaction with protein 4.1, a protein that connects spectrin to the actin cytoskeleton in dendritic spines. Mimicking CASK sumoylation dramatically impairs spine formation[53]. According to the importance of sumoylation in the post-synaptic formation and maturation, our findings demonstrate a role of sumoylation in spine elimination and maturation by tuning FMRP dimerization within dendritic mRNA granules. Altogether, these data shed light on the role of sumoylation as a critical molecular regulator in neuronal development and maturation.

Interestingly, we demonstrate that the sumoylation of FMRP is triggered upon mGlu5R activation. mGlu5R has been previously reported to differentially regulate FMRP function depending on its subcellular localization. For instance, a direct involvement of FMRP was shown in targeting and transport of several mRNAs from the soma along dendrites upon mGlu5R activation[52]. Furthermore, the repression of mRNA translation exerted by FMRP in dendrites is counteracted by the activation of mGlu5R[51]. Here, we unravel a novel activity-dependent regulation of the FMRP function. We show that mGlu5R-induced sumoylation of FMRP drives its own dissociation from dendritic mRNA granules to regulate both spine elimination and maturation.

It has been previously described that FMRP is a target of mGluR-dependent PTMs[11,13,14,54,55]. Activation of mGluRs in neurons induces a rapid dephosphorylation of FMRP C-terminal region as a result of an enhanced protein phosphatase 2A (PP2A) activity[11]. Conversely, mGluR activation that lasts longer than 5 min results in an mTOR-mediated PP2A suppression followed by rapid rephosphorylation of FMRP C-terminus by the ribosomal protein S6 kinase (S6K1)[11,55]. Accordingly to the role of phosphorylation in controlling FMRP function, the lack of S6K1-dependent FMRP phosphorylation mimics FMRP loss of function and leads to an increased expression of the FMRP target mRNA SAPAP3[55]. In addition, Nalavadi et al.[14] described a rapid ubiquitination of the C-terminal part of FMRP upon stimulation with the mGlu5R agonist DHPG in rat cultured neurons. FMRP ubiquitination promotes a proteasome-mediated FMRP degradation, which in turn controls FMRP levels at the synapse. Interestingly, these authors showed that FMRP ubiquitination requires a prior FMRP-dephosphorylation carried by PP2A. Taken together, these pieces of evidence suggest a crosstalk between various PTMs in the regulation of FMRP function. Here, we demonstrate that mGlu5R activation triggers a rapid sumoylation of FMRP. This event promotes the release of FMRP from transport mRNA granules. Thus, the present study adds another level of complexity to the post-translational regulation of FMRP and advances our understanding of the activity-dependent control of FMRP function in neurons. It will therefore be of future

interest to examine whether the interplay between these PTMs could take place to orchestrate the mGlu5R-dependent regulation of FMRP.

The present study shows that the activation of mGlu5R directly promotes FMRP sumoylation, regulating its neuronal function in spine elimination and maturation. Our work therefore raises the intriguing possibility that the impairment of FMRP sumoylation could contribute to FXS physiopathology. Recent publications have reported missense point mutations within the *FMR1* gene in patients affected by FXS. Importantly, these mutations lead to amino-acid changes close to the SUMO active sites of FMRP (F126S[56] and R138Q[57]). Similarly to our data on the K88,130R FMRP mutant, the FXS R138Q mutation does not modify the expression of FMRP nor its RNA-binding properties, indicating that the pathogenicity is caused independently of the FMRP expression level and the ability of FMRP to bind mRNAs[58]. To date, no data have been reported regarding the functional impairment due to the F126S mutation. Our data report that the reintroduction of the FMRP WT but not the K88,130R mutant in $Fmr1^{-/y}$ neurons promotes spine maturation and elimination demonstrating that FMRP sumoylation is critical for these processes. Therefore, an interesting possible explanation could be that the F126S and R138Q FXS mutations, which are very close to the active K130-SUMO site, would directly impact on the mGlu5R-dependent regulation of FMRP sumoylation and consequently, on post-synaptic FMRP-driven regulatory events. Future work will have to be performed aiming at understanding the effect of these FXS mutations on FMRP sumoylation. These next exciting steps will allow assessing whether FMRP sumoylation defects participate in the pathophysiology of FXS patients, raising the possibility to identify new targets and potentially develop novel therapeutic approaches.

## Methods

**Constructs**. GFP-FMRP was obtained by subcloning the isoform 1 of the human FMR1 sequence into the EcoR1/Pst1 site of the mammalian expression vector pEGFP-C2 (Clontech). GFP-/Dendra2-/GST-/His-FMRP mutant constructs were all made by site-directed mutagenesis using the Quick-change mutagenesis solution (Agilent). pSinRep5 constructs used to produce Sindbis particles were generated using the Gateway recombination technology (Invitrogen). All constructs were then entirely sequenced.

**Building model for FMRP-SUMO1**. Three X-ray structures of human FMRP are available in Protein Data Bank (PDB, http://www.rcsb.org; PDB ID: 4OVA residues 1–209 at 3.0 Å resolution[59], 4QVZ residues 1–213 at 3.2 Å resolution, and 4QW2 residues 1–213 with the mutation R138Q at 3.0 Å resolution[60]). The solvent ASA values for each residue have been calculated using Naccess tool[61] on all monomers of each PDB files (4 for 4OVA, 2 for 4QVZ, and 2 for 4QW2). We calculated the average values for K88 and K130 for each structure. The classical parameters used are 1.4 for the radius of the "solvent" sphere and 25% for the threshold that determines if a residue is considered as buried or exposed. We utilized the X-ray structures of human FMRP PDB ID: 4OVA residues 1–209 at 3.0 Å resolution[59] and of human SUMO1 PDB ID: 4WJQ at 1.35 Å resolution[62]. To build models of FMRP modified with the SUMO1 protein, we first verified the shape compatibility and then used the Pymol software to manipulate the structures, make and visualize the FMRP-SUMO1 models.

**Mouse lines and rat strain**. All animals (3–10-month-old pregnant female Wistar rats from Janvier, St Berthevin, France; 3–10 month-old female C57BL/6 WT and *Fmr1* knockout *(Fmr1$^{-/y}$)* mice[10]) were handled in our facility in accordance with the European Council Guidelines for the Care and Use of Laboratory animals and approved by the Animal Care and Ethics Committee (Comité Institutionnel d'Ethique Pour l'Animal de Laboratoire N°28, Nice, France; project reference NCE/2012-63). All animals had free access to water and food. The light cycle was controlled as 12 h light and dark cycle and the temperature was maintained at 23 ± 1 °C. Protocols to prepare primary neuronal cultures from mouse embryos at E15.5 or at E18 for rats were also approved by the Animal Care and Ethics Committee (Comité Institutionnel d'Ethique Pour l'Animal de Laboratoire N°28, Nice, France; project reference NCE/2012-63). All mice were maintained on a C57BL/6 genetic background, whereas Wistar rats were exclusively from a commercial source (Janvier). The *Fmr1* knockout *(Fmr1$^{-/y}$)* mouse line[10] was maintained on a C57BL/6 background.

**Mouse and rat brain lysate preparation**. Brain lysates were prepared as previously described[26] from post-natal P1–3 mouse or P5–7 rat brains. Briefly, freshly dissected brains were transfered in 5 volumes (w/v) of ice-cold sucrose buffer (10 mM Tris-HCl, pH 7.4, 0.32 M sucrose) supplemented with a protease inhibitor cocktail (Sigma, 1/100), Pefabloc 0.5 mM (Roche), MG132 100 μm (Enzo), ALLN 100 μm (Sigma), and 20 mM freshly prepared NEM (Sigma), and homogenized at 4 °C using a Teflon-glass potter and a motor-driven pestle at 500 rpm. Nuclear fraction and cell debris were pelleted by centrifugation at 1000×g for 10 min. The post-nuclear S1 fraction (supernatant) was collected and protein concentration measured using the BCA protein assay (Bio-Rad).

**Primary neuronal cultures**. Hippocampal and cortical neurons were prepared from embryonic (E18) pregnant Wistar rats as previously described[26] or from WT or $Fmr1^{-/y}$ E15.5 pregnant C57BL/6 mice. Briefly, neurons were plated in Neurobasal medium (Invitrogen, France) supplemented with 2% B27 (Invitrogen), 0.5 mM glutamine and penicillin/streptomycin (Ozyme) on 60-mm dishes or 24-mm glass coverslips (VWR) pre-coated with poly-L-lysine (0.5 mg mL$^{-1}$; Sigma). Neurons (800,000 cells per 60-mm dish or 110,000 cells per coverslip) were then fed once a week with neurobasal medium supplemented with 2% B27 and penicillin/streptomycin for a maximum of 3 weeks.

**Cell transfection**. COS7 cells and primary neurons (14–16 DIV) were transfected using Lipofectamine 2000 (Invitrogen) according to the manufacturer's instructions and used 48–72 h post transfection.

**Sindbis virus production and neuronal transduction**. Attenuated Sindbis viral particles (SINrep(nsP2S726)) were prepared and used as previously described[38–40]. Briefly, cRNAs were generated from the pSinRep5 plasmid containing the sequence coding for WT or mutated GFP-FMRP constructs and from the defective helper (pDH-BB) plasmid using the Mmessage Mmachine SP6 solution (Ambion). cRNAs were then mixed and electroporated into BHK21 cells. Pseudovirions present in the culture medium were collected 48 h after electroporation and concentrated using ultracentrifugation on SW41Ti. Aliquots of resuspended Sindbis particles were then stored at −80 °C until use. Neurons were transduced at a multiplicity of infection (MOI) of 0.1–2 and returned to the incubator at 37 °C under 5% CO$_2$ for 24–30 h depending on their subsequent utilization.

**Bacterial sumoylation assay in *Escherichia coli***. Bacterial sumoylation assays were performed as previously described[31,35]. Briefly, competent *E. coli* BL21(DE3) cells (Invitrogen, France) expressing pE1-E2SUMO1 were transformed with 1 μg of pET-expression plasmid (Novagen) to express the WT or non-sumoylatable forms of His-tagged FMRP were selected on LB-Agar plates containing chloramphenicol (50 μg mL$^{-1}$) and ampicillin (50 μg mL$^{-1}$). A 10 mL preculture was then used to inoculate 50 mL of LB containing chloramphenicol and ampicillin. After incubation under shaking at 37 °C until OD$_{600}$ reaches 0.7, cells were cooled down to 20 °C and isopropyl-β-D-thiogalactopyranoside (IPTG) was added at a concentration of 1 mM. After 4 h at 20 °C, bacteria were pelleted by centrifugation at 4 °C at 7000×g and kept at −80 °C until use. Pellets were resuspended in 1 mL lysis buffer (25 mM Tris pH 8, 300 mM KCl, 1 mM EDTA, 20% glycerol, 5% ethanol, 0.5% NP40, 0.5 M urea, 1 mM DTT) supplemented with proteases inhibitors (leupeptine 1 μg mL$^{-1}$, Pepstatine 1 μg mL$^{-1}$, Aprotinine 1 μg mL$^{-1}$, Pefabloc 0.5 mM, and freshly prepared NEM 20 mM), and incubated under rotation for 30 min at 4 °C in the presence of 5 mg mL$^{-1}$ lysozyme. Bacterial cytoplasmic membranes were then solubilised by addition of 1 mg mL$^{-1}$ sodium deoxycholate and released DNA digested by incubation with 50 μg mL$^{-1}$ of DNAse I and 10 mM MgCl$_2$ for 30 min at 4 °C. Cellular debris were pelleted by centrifugation at 20,000×g for 15 min at 4 °C and supernatants were incubated with 40 μL of nickel agarose beads (Qiagen) for 2 h at 4 °C under gentle rotation. After three washes (25 mM Tris pH 8, 50 mM KCl, 1 mM EDTA, 20% glycerol, 0.1% Triton X-100, 0.5 M urea, 1 mM DTT), purified proteins were eluted in 200 μL of βME-reducing sample buffer for 5 min at 95 °C.

**COS7 sumoylation assay**. Mycoplasm-free COS7 cells (ATCC reference CRL-1651, Molsheim, France) at 60% of confluence in six-well plates were co-transfected using 1 μg of the eukaryotic expression vector pTL1-FMRP plasmid[63] or its derived non-sumoylatable mutants with 0.5 μg of mCherry or mCherry-SUMO1 plasmids[26] and 0.5 μg of plasmid coding for Flag-Ubc9 using Lipofectamine 2000 (Invitrogen) according to the manufacturer's instructions. After 48 h of expression, cells were washed once in PBS containing 20 mM NEM and reduced for 5 min at 95 °C in βME-containing sample buffer.

**CLIP analysis**. To isolate neuronal mRNAs associated with WT and SUMO-deficient GFP-FMRP mutant, UV cross-linking, and FMRP immunoprecipitations were performed on 20 DIV $Fmr1^{-/y}$ neurons transduced (MOI of 3) at day 19 to express free GFP, the WT, or the non-sumoylatable K88,130,614R form of GFP-FMRP. RNAs and proteins were cross-linked through three rounds of UV irradiation (400 mJ each; 254 nm). Cells were then scraped in ice-cold PBS, collected by centrifugation, and lysed in NP40 buffer as described in ref.[64]. For each assay, 5 μg of affinity-purified rabbit anti-FMRP antibody (Ab#056) was used to immunoprecipitate 1 mg of neuronal extracts and 2% of the lysate was used for assessment of relative RNA expression in the input material. IPs were then carried out at 4 °C for 4 h and 2% of the homogenate and 10% of the immunoprecipitates were saved to check for the IP quality using anti-FMRP immunoblots. After three washes in lysis buffer (50 mM HEPES, pH 7.4, 150 mM NaCl, 0.5% NP40, 10 mM EDTA, 1 mM NaF, 0.5 mM DTT, protease and phosphatase inhibitors (Pierce), proteins were digested with proteinase K (1 μg mL$^{-1}$) for 30 min at 56 °C. IP and input RNAs were purified through two successive rounds of phenol/chloroform extraction, then reverse transcribed using a mix of Oligo dT and random primers and Superscript II enzyme (Invitrogen) according to the manufacturer's protocol. RT reactions were diluted two times and 1 μL of diluted material was used for qPCR analysis. Relative enrichment of the amplified RNA in the IP vs the input in each condition was calculated with the $2^{-deltaCt}$ (Ct$_{IP}$−Ct$_{input}$).

Oligonucleotides (5′–3′) used in RNA work were as follows: *Fmr1*_F: GAACAAAAGACAGCATCGCT; *Fmr1*_R: CCAATTTGTCGCAACTGCTC; *Camk2a*_F: TATCCGCATCACTCAGTAC; *Camk2a*_R: GAAGTGGACGATCTGCCATTT; *Sapap3*_F: ACCATGTAACCCCGGCTG; *Sapap3*_R: CCTTGATGTCAGGATCCCC; *Fxr1*_F: GTGCAGGGTCCCGAGGT; *Fxr1*_R: GGTGGTGGTAATCGGACTTC; *Kif3c*_F: GGTCCCATCCCAGATACAGA; *Kif3c*_R: CCAGAAAGCTGTCAAACCTC; *Tubb3*_F: CGAGACCTACTGCATCGACA; *Tubb3*_R: CATTGAGCTGACCAGGGAAT; *PP2a*_F: GTCAAGAGCCTCTGCGAGAA; *PP2a*_R: GCCCATGTACATCTCCACAC; β-*actin*_F: ACGGCCAGGTCATCACTATTG; β-*actin*_R: CACAGGATTCCATACCCAAGA; *PSD95*_F: GGCGGAGAGGAACTTGTCC; *PSD95*_R: AGAATTGGCCTTGAGGGAGGA; *Map1b*_F: TCCGATCGTGGGACACAAACCTG; *Map1b*_R: AGCACCAGCAGTTTATGGCGGG.

**Immunoprecipitation**. Proteins from rodent brain lysates or cultured neurons were solubilized for 1 h at 4 °C under gentle rotation in lysis buffer (10 mM Tris-HCl, pH 7.5, 10 mM EDTA, 150 mM NaCl, 1% Triton X-100, 0.1% SDS) supplemented with a protease inhibitor cocktail (Sigma, 1/100), Pefabloc 0.5 mM (Roche), MG132 100 μm (Enzo), ALLN 100 μm (Sigma), and 20 mM freshly prepared NEM (Sigma). Then, NaCl concentration was raised to 400 mM and lysates were sonicated for 10 s, further incubated for 30 min at 4 °C and clarified (for primary neuronal extracts) or not (for brain homogenates) at 20,000×g at 4 °C for 15 min. Supernatants were diluted 2.5-fold with lysis buffer devoid of NaCl and pre-cleared for 1 h with a 50/50 mix of untreated and pre-blocked protein G-sepharose beads (Sigma) with a blocking buffer (PBS containing 5 mg mL$^{-1}$ BSA, 5 mg mL$^{-1}$ Dextran (40 kDa), 1 mg mL$^{-1}$ gelatin, yeast t-RNA 0.1 mg mL$^{-1}$, and glycogen 0.1 mg mL$^{-1}$) for 1 h at 4 °C. Proteins (800 μg) from pre-cleared lysates were incubated with either 8 μg of mouse monoclonal anti-SUMO1 antibody (Ab#D11, Santa-Cruz), 4 μg of custom rabbit anti-FMRP (Ab#056, Supplementary Fig. 1), or 12 μg commercially available rabbit anti-FMRP (#Ab17722, Abcam; Supplementary Fig. 1) antibodies (or their corresponding IgGs as IP control) for 1 h at 4 °C and then overnight at 4 °C with 30 μL of pre-blocked protein G-sepharose beads (Sigma). Precipitates were washed three times with 1 mL lysis buffer and proteins were eluted by boiling the beads 5 min in βME-reducing sample buffer before SDS-PAGE.

**Immunoblotting**. Protein extracts were resolved by SDS-PAGE, transferred onto nitrocellulose membrane (Hybond-C Extra, Amersham or BioTraceNT, PALL), immunoblotted with the indicated concentration of primary antibodies and revealed using the appropriate horseradish peroxidase (HRP)-conjugated secondary antibodies (GE healthcare) or True Blot (Rockland, Tebu-Bio). Proteins were then identified using Immobilon Western (Millipore) or Western Lightning Ultra (Perkin Helmer) chemiluminescent solutions and images acquired on a Fusion FX7 system (Vilber Lourmat). Full-size blots for cropped gels can be found in Supplementary figures 6, 7.

**Immunocytochemistry**. Neurons (18–21 DIV) were fixed in phosphate-buffered saline (PBS) containing 3.7% formaldehyde and 5% sucrose for 1 h at room temperature (RT). Neurons were then permeabilized for 20 min in PBS containing 0.1% Triton X-100 and 10% horse serum (HS) at RT and immunostained with either a rabbit monoclonal anti-S6 (1/200; Cell Signaling), a goat anti-Staufen1 (1/100; Santa-Cruz), a goat anti-Staufen2 (1/100; Santa-Cruz), a rabbit anti-FXR1 (1/100[65]), a mouse monoclonal anti-Ubc9 (1/50; BD Bioscience, France), a mouse anti-SUMO1 (1/50; Ab#D11, Santa-Cruz; 1/50 Ab#2F5–1, DSHB) or rabbit anti-FMRP (1/200; Custom Ab#056 or 1/50; Ab#4317s, Cell Signaling) antibodies in PBS containing 0.05% Triton X-100 and 5% HS. Cells were washed three times in PBS and incubated with the appropriate secondary antibodies (1/400) conjugated to Alexa488 or Alexa594, and mounted with Mowiol (Sigma) until confocal examination.

**Ratiometric calcium imaging**. Mouse cortical/hippocampal neurons (19–23 DIV) were loaded in neurobasal containing 20 μm Fura-2AM (Invitrogen) for 30 min. After two washes in physiological 1.6 mM calcium-containing buffer (139 mM NaCl, 1.25 mM glucose, 15 mM Na$_2$HPO$_4$, 1.8 mM MgSO$_4$, 1.6 mM CaCl$_2$, 3 mM

KCl, 10 mM HEPES), Fura-2AM-loaded neurons were imaged at 37 °C on an inverted AxioObserver microscope (Carl Zeiss) equipped with a 300 W Xenon lamp (Suttler instruments) and a Fluar 40× (numerical aperture (NA) 1.4) oil immersion objective. Fura-2AM was sequentially excited at 340 and 380 nm and the emission monitored at 510 nm. Images were acquired with a cascade 512 EMCCD camera every 2 s and digitized using Metafluor software (Roper scientific). The intracellular calcium concentration was estimated by measuring the F340/380 nm ratio of fluorescence. Neurons were treated for 40 s with 100 μM DHPG in 1.6 mM calcium-containing buffer.

**GFP-FMRP-associated granules analysis.** *Fmr1*$^{-/y}$ neurons were co-transfected to express mCherry with the WT or the non-sumoylatable mutant form of GFP-FMRP either for 48 or 72 h. Cells were then rinse twice in PBS and fixed in PBS containing 3.7% formaldehyde and 5% sucrose for 1 h at RT and mounted with Mowiol until use.

**smFISH assays.** smFISH assays were performed as described previously in ref. [44]. Briefly, *Fmr1*$^{-/y}$ neurons grown on glass coverslips were transfected as above at 12 DIV to express the WT or the K88,130R mutant form of GFP-FMRP for 48–60 h. Cells were then fixed in PBS containing 4% formaldehyde for 10 min at RT. smFISH assays were performed as described previously in ref. [44] with the following modified prehybridization buffer: formamide 10%, NaCl 68.5 mM, KCl 1.35 mM, KH$_2$PO$_4$ 1 mM, Na$_2$HPO$_4$ 5 mM, SSC 2× (Euromedex), dextran sulfate 10% (Sigma), ribonucleoside vanadyl complexes 10 mM (Sigma), BSA 2 mg mL$^{-1}$, salmon sperm DNA 0.67 mg mL$^{-1}$ (Sigma), yeast tRNA 0.67 mg mL$^{-1}$ (Sigma). Neurons were incubated overnight at 37 °C in the presence of GFP Quasar 570-labeled, PSD-95 or CamKII Quasar 670-labeled Stellaris probes (12.5 picomoles in 100 μL of prehybridization buffer), washed 2 times with pre-warmed 10% formamide in 2× SSC for 20 min at 37 °C, three times 1 min with 2× SSC, and twice for 5 min with 2× SSC under mild agitation prior to coverslip mounting in Vectashield (Cliniciences).

GFP Stellaris probes (used to detect GFP-FMRP mRNA) labeled with Quasar 570 dye were (5′–3′) as follows: tcctcgcccttgctcaccat, atgggcaccaccccggtgaa, gtcgccgtccagctcgacca, cgctgaacttgtggccgttt, tcgccctcgccctcgccgga, ggtcagcttgccgtaggtgg, cggtggtgcagatgaacttc, ggccagggcacgggcagctt, taggtcagggtggtcacgag, tagcggctgaagcactgcac, gtgctgcttcatgtggtcgg, gcatggcggacttgaagaag, cgctcctggacgtagccttc, gtcgtccttgaagaagatgg, tcggcgcgggtcttgtagtt, ggtgtcgccctcgaacttca, ttcagctcgatgcggttcac, gtcctccttgaagtcgatgc, agcttgtgcccaggatgtt, gtggctgttgtagttgtact, ttgtcggccatgatatagac, caccttgatgccgttcttct, atgttgtggcggatcttgaa, gagctgcacgctgccgtcct, tgttctgctggtagtggtcg, agcacggggcctcgccgat, caggtagtggttgtcgggca, ttgctcagggcggactgggt, atcgcgctctcgttggggt, cgaactccagcaggaccatg, agagtgatcccggccggcggt, cttgtacagctcgtccatgc.

PSD-95 Stellaris probes, labeled with Quasar 670 dye were (5′–3′) as follows: ctctatgatcttctcagctg, taggcccttttgataagcttg, tgcgatgctgaagccaagtc, ctattatctccagggatgtg, ccttcgatgatcttggttac, aggatcttgtctccgatctg, tcatgcatgacatcctctag, atatgtgttcttcagggctg, ccaccttttaggtacacaacg, catagctgtcactcaggtag, tatgaggtttgatgtctggg, tagctgctatgactgatctc, tcaacaccattgaccgacag, tgttcatgactggcattgcg, tactgagcgatgatcgtgac, cgaatcggctatactcttct, ataagctgttcccgaagatc, tgatatagaagcccccgcttg, ttgtcgtagtcaaacagggc, tcaagaaaccgcagtccttg, gctggcgtcaattacatgaa, catcggtctcactgtcagag, tttgctgggaatgaagccaa, tctcatagctcagaaccgag, aaggatgatgatgggcgag, agaagatcatcgttggcacg, aaacttgtcggggaactcgg, tcgtatgagggacacaggat, tatctcatattcccgcttag, cgggaggagacaaagtggta, tgaatgtccttctccattttt, cagcctcaatgaacttgtgc, tagaggtggctgttgtactg, gcattggctgagacatcaag, ggatgaagatggcgataggg, cgcttattgatctctagcac, agatctcttcaaagctgtcg, cttcgatgcacgtttcact.

CaMKII Stellaris probes, labeled with Quasar 670 dye were (5′–3′) as follows: tactcttctgtgaatcgggt, taatcttggcagcatactcc, cttcaacaagcggcagatgc, tcatggagtcggacgatatt, accagtaaccagatcgaaga, ggccacaatgtcttcaaacag, ggcatcagcttcactgtaat, tccaagatctgctggataca, catctggtgacagtgtagca, ttcacagcagcgcccttgag, tatccaggtgtccctgcaag, cttcctcagcacttctgggg, aggtccacgggcttcccgta, agatatacaggatgacgcca, ctggtcttcatcccagaacg, ctttgatctgctggtacagg, gatgggaaatcataggcacc, ggtgacggtgtcccattctg, tgatcagatccttggcttct, gggttgatggtcagcatctt, tcagcggccgtgatgcgttt, agatccatgggtgcttgaga, tctcctgtctgtgcatgcag, ctccggagaagttcctggtg, gtgctctcagaagattcctt, tcttcgtcctccaatggtggt, ttcctgtttgcgcactttgg, gctgctctgtcactttgata, ccattgcttatggcttcgat, cttcgtgtaggactcaaagt, ctgtcattccagggtcgcac, cccagggcctctggttcaaa, gaatcgatgaaagtccaggc, gggaccacaggttttcaaaa, tgactcgtcacccatcaggt, atgcggatataggcgatgca, gcctgcatccaggtactgag, cagacgcgggtctcctctga, atctgtggaagtggacgatc, cgagtacataggtggcaatg, aaatacacggaagtttggct, agatgtccgttaacgcaaaa, acagcattccatacaagagc, tatagctcacatgtaggcga, ctgagccttatgaagaagcc, ggattgtagatcctgcatgg, catggagcttgtcagatgag, tttgagcagtggtcattcaa.

**Analysis of spine morphology.** *Fmr1*$^{-/y}$ or WT neurons were transduced at 18 DIV with Sindbis virus (MOI of 1) expressing free GFP, the WT or mutated forms of GFP-FMRP for either 24 h (*Fmr1*$^{-/y}$ neurons), or 30 h (WT neurons) before use.

Cells were then fixed using PBS containing 3.7% formaldehyde and 5% sucrose for 1 h at RT and mounted in Mowiol before confocal examination.

**Confocal imaging.** For fixed cells, confocal images (1024 × 1024) were acquired with a ×63 oil immersion lens (numerical aperture NA 1.4) on an inverted TCS-SP5 confocal microscope (Leica Microsystems, Nanterre, France). Z-series of 6–8 images of randomly selected secondary dendrites were compressed into two dimensions using the maximum projection of the LASAF acquisition software (Leica). Manders' co-localization parameters were computed using the JaCoP plug-in from the ImageJ software[66] when required.

For GFP-FMRP-containing granule measurements, two Z-series were acquired. The first was acquired at low laser intensity to clearly identify large granules without any pixel saturation and the second series was recorded at a higher laser intensity to detect smaller granules. These two Z-series were then averaged and compressed into two dimensions by a maximal projection. Measurements of the surface of GFP-FMRP-containing granules along dendrites were determined automatically using an home-made ImageJ macro program. Briefly, granules and dendrites were segmented in each image, and the length of the dendritic tree was measured after a step of skeletonization. The data were then imported in GraphPad Prism software for statistical analysis.

For dendritic spine imaging, Z-series of six to eight images of secondary dendrites from GFP-expressing neurons were compressed into two dimensions by a maximal projection using the LASAF software. About 3000–4500 spines were analyzed per condition (two to four dendrites per neuron and from 20 to 30 neurons per condition from four independent experiments, which were done blind for two of them). At the time of acquisition, laser power was adjusted so that all spines were below the saturation threshold. To analyze dendritic protrusions, projection images were imported into NeuronStudio software[67], which allows for the automated detection of immature and mature dendritic spines. The length of individual spines was automatically measured and data were imported in GraphPad Prism software for statistical analysis. Mature spines were characterized by a head diameter ranging from 0.3 to 1 μm and a spine length between 0.4 and 3 μm. Immature spines corresponded to protrusions with a head diameter below 0.3 μm and a spine length ranging from 0.5 to 6 μm.

**Fluorescence lifetime imaging experiments.** *Fmr1*$^{-/y}$ neurons co-expressing mCherry with the WT or the non-sumoylatable mutant form of GFP-FMRP for 72 h were fixed in PBS containing 3.7% formaldehyde and 5% sucrose for 1 h at RT and mounted using Mowiol. FLIM was then performed on a Nikon A1R confocal laser-scanning microscope equipped with time-correlated single-photon counting electronics (PicoHarp 300; PicoQuant). Excitation was obtained using a pulsed laser LDH-D-C-485 (PicoQuant) at a repetition rate of 40 MHz allowing the acquisition of the full intensity decay. Fluorescence emission was collected by a hybrid photomultiplier detector (PicoQuant) through a 60 × λ*S*, NA 1.4, oil objective (Nikon Instruments) and band-pass filter (520/35). The following parameters were kept constant for all acquisition: pixel size (70 nm, 512 × 512), pixel dwell time (4.8 μs), and acquisition time (5 min per image). So as to limit pile-up and to accumulate enough photons within the 5 min acquisition time, laser excitation power was adjusted to obtain a count rate between 0.4 and 2 MHz[68]. In these conditions, there was no measurable photobleaching. Each field of view was also acquired in conventional confocal mode. EGFP and mCherry channels were, respectively, acquired using the 488-nm excitation with the 525/50-nm band-pass detection and the 561-nm excitation with the 595/50-nm band-pass detection.

Fluorescence lifetime was measured by fitting the intensity decays with a monoexponential decay model reconvolved with experimental IRF (instrument response function) using the software SymPhoTime (PicoQuant). Intensity decays were fitted pixel by pixel to provide FLIM images and calculated lifetimes represented using a pseudo-color scale ranging from 1.7 to 2.2 ns. To improve robustness of the fit, IRF parameters are fixed as the IRF is expected to be invariant over the acquisition field. The robustness of the fit was assessed by the calculated standard weighted least square ($X^2$) and the residual[69]. Values of the reduced $X^2$ should be close to 1 and residue should be randomly distributed around zero. The average lifetime of the FLIM image (Tau, ns) was determined from the barycentre of the frequency histogram associated with the FLIM image. To calculate the fluorescence lifetime of individual granules, the intensity decay resulting from all the photons of the granule was fitted using a monoexponential model reconvolved with IRF.

To get enough photons at each pixel for an accurate intensity decay fit, only granules with >10,000 photons (integrated number of photons over the decay) were analyzed, with a minimum pixel threshold of 500 counts for background rejection. To reach 10,000 photons per granules and to reject granules, which were largely out of focus, only granules >0.35 μm$^2$ were analyzed (segmentation using ImageJ). To avoid pulse pile-up and to collect photons fast enough to meet the above criteria, count rate was kept between 0.4 and 4 MHz. Clusters with higher count rate were excluded from the analysis. In those conditions, the fluorescent lifetime was invariant.

**Dendra2-FMRP-containing granule photoconversion experiments.** Experiments were performed as previously described[26,48]. Briefly, live *Fmr1*$^{-/y}$ neurons

expressing the WT or mutated Dendra2-FMRP from were kept in Earle's buffer (25 mM HEPES-Tris, pH 7.4, 140 mM NaCl, 5 mM KCl, 1.8 mM CaCl$_2$, 0.8 mM MgCl$_2$, 0.9 g L$^{-1}$ glucose) on the heated stage (set at 37 °C) of a Nikon Ti inverted microscope and imaged using an ultraview spinning disk confocal system (Perkin Elmer, France). Cells were then stimulated or not with 50 μm DHPG in Earle's buffer. After a 10-min incubation time in either control or DHPG solution, Dendra2-FMRP-granules were photoconverted through a ×100/ NA 1.4 oil immersion objective for 30 ms using 405-nm laser light (50 mW, 15%). The red photoconverted Dendra2-FMRP was excited using a 561-nm laser light (50 mW, 17%) and two-dimensional-time series (2 Hz) were collected for 10 min. The decrease in red fluorescence from the Dendra2-FMRP photoconverted granules was measured over time using Velocity 6.3 software and data expressed as the percentage of the initial red photoconverted fluorescence ($F/F_0$). Curves were fitted using a monoexponential decay equation and data analyzed using GraphPad Prism.

**GST- and His-FMRP production and purification.** GST- or His-FMRP (1–160) proteins were produced in *E. coli* BL21(DE3) cells (Invitrogen, France). A single colony was picked and used to inoculate 25 mL of LB broth supplemented with 50 μg mL$^{-1}$ ampicillin. This was used to inoculate 500 mL of LB and was shaken at 37 °C until OD$_{600}$ reached 0.8. Cells were then transferred at 20 °C and protein synthesis induced by addition of 1 mM IPTG (Sigma, France). After 4 h at 20 °C, cells were pelleted by centrifugation at 7000×*g* for 5 min and then gently resuspended in ice-cold PBS and frozen at −80 °C until use. Pellets were then resuspended in 5 mL lysis buffer (25 mM Tris-HCl pH 8, 300 mM KCl, 1 mM EDTA, 20% glycerol, 5% ETOH, 0.5% NP40, 0.5 M urea) supplemented with 1% protease inhibitor cocktail (Sigma-Aldrich, France). Cells were disrupted by incubation with 1% lysozyme (Sigma, France) for 30 min at 4 °C followed by another 30 min in the presence of 0.1% deoxycholic acid, 10 mM MgCl$_2$ and 200 ng μL$^{-1}$ DNase. Lysates were then clarified by centrifugation at 10,000×*g* for 15 min. GST- or His-tagged proteins were purified using either glutathione gel (GE Healthcare) for GST- and GST-FMRP or Nickel resin (Qiagen) for His-fusion proteins. Proteins were then concentrated on Amicon 3-kDa cutoff filters (Millipore) by centrifugation and resuspended in PBS. Concentrations of purified proteins were determined using the BCA protein assay (Bio-Rad) and protein quality assessed by SDS-PAGE and Coomassie Blue protein staining (Clinisciences).

**GST-FMRP/His-FMRP dimerization.** GST- (control) or GST-FMRP (1–160) fusion proteins (1 μg) were incubated with an excess of 2 μg His-FMRP (1–160) for 2 h at 4 °C in dimerization buffer (50 mM Tris-HCl pH 8, 150 mM NaCl, 2.5 mM MgCl$_2$, 0.5% NP40, 0.5 mM DTT, 1% protease inhibitor cocktail) to allow for GST-FMRP/His-FMRP dimerization. Then, 50 μL of glutathione beads (GE Healthcare) were added to the dimerization mix and incubated at 4 °C for 2 h. After five washes in dimerization buffer at 4 °C, immobilized GST-FMRP (1–160)—His-FMRP (1–160) dimers were processed for in vitro sumoylation assays.

**In vitro SUMO assays.** Immobilized GST-FMRP/His-FMRP dimers were incubated with 0.15 μg of E1-activating complex (Enzo Life science), 0.1 μg of E2 Ubc9 (Enzo Life science), 3 μg of SUMO1-GG in 20 μL of in vitro SUMO reaction mix (20 mM HEPES pH 7.3, 110 mM KOAc, 2 mM Mg(OAc)$_2$, 0.5 mM EGTA, 1 mM DTT 0.05% Tween 20, 0.2 mg mL$^{-1}$ ovalbumin) including the ATP regenerating system (20 mM ATP, 10 mM creatine phosphate, 3.5 U mL$^{-1}$ of creatine kinase, and 0.6 U mL$^{-1}$ of inorganic pyrophosphatase (Sigma-Aldrich) for 2 h at 30 °C). After centrifugation for 5 min at 3000×*g* at 4 °C, the supernatant containing the released His-FMRP (1–160) and the pellet containing the remaining immobilized GST-FMRP/His-FMRP dimers were denatured at 95 °C for 10 min in 5× Laemmli buffer containing 7.5% β-mercaptoethanol and analyzed by immunoblotting with FMRP #2F5-1 antibodies.

**Electrophysiological recordings.** Patch clamp experiments were carried out at RT (22–25 °C) on mixed cultured cortical/hippocampal neurons obtained from *FMRP*$^{-/y}$ mice (four different cultures). *FMRP*$^{-/y}$ neurons (18 DIV) were transduced for 24–26 h with attenuated Sindbis virus to express GFP-FMRP WT or the non-sumoylatable GFP-FMRP-K88,130,614R. Patch pipettes displayed a resistance of 4–7 MΩ and filled with a solution containing (in mM): 2 Na$_2$-ATP, 130 CsMeSO$_4$, 5 CsCl, 2.5 MgCl$_2$, 1 Na-GTP, 5 EGTA, and 10 HEPES (pH adjusted to 7.2 with CsOH). The extracellular bathing solution contained (in mM): 145 NaCl, 5 KCl, 2 CaCl$_2$, 2 MgCl$_2$, 10 HEPES, 10 glucose, 0.02 bicuculline, and 0.00025 TTX (pH adjusted to 7.4 with NaOH). We used the whole-cell configuration to record mEPSCs from GFP-positive neurons that were voltage-clamped at −70 mV, i.e., the estimated reversal potential for chloride. mEPSCs were recorded for 10 min, starting 1–2 min after the whole-cell mode was achieved and series resistances were monitored every 50 s by injecting a 5 mV hyperpolarizing current for 10 ms. Data were sampled at 20 kHz, low-pass filtered at 5 kHz (Axopatch 200B Molecular Devices), digitalized (Digidata 1440, Molecular Devices) and recorded using Clampex software (pClamp 10, Molecular Devices). Analysis of series resistances and mEPSCs were performed offline using Clampfit software (pClamp 10, Molecular Devices). mEPSCs were analyzed over periods of 200 s for which series resistances were stable, i.e., did not vary for >25%.

**Data and statistical analysis.** Statistical analyses were calculated using GraphPad Prism (GraphPad software, Inc). All data are expressed as mean ± s.e.m. Unpaired *t* test (Fig. 3i) or non-parametric Mann–Whitney test (Figs. 4c and 5d, f, g) were used to compare medians of two data sets. For spine morphogenesis experiments, values represent means ± s.e.m. Statistical significance for multiple comparison data sets was computed using a one-way analysis of variance with a Bonferroni post-test (Figs. 2b–d, 5e, 7b–d, and Supplementary Fig. 2b–d). Normality for all groups was verified using the Shapiro–Wilk test. According to the Levene variance test, variances were homogenous for the percentage of immature and mature spines (Figs. 2b–d, 7b–d, and Supplementary Fig. 2b–d). For FLIM, data distributions were represented as box and whiskers plots displaying upper and lower quartiles, and maximum and minimum values in addition to median. For electro-physiological data, distributions were analyzed by a Kolmogorov–Smirnov test (Supplementary Fig. 3b, c). *$p < 0.05$ was considered significant.

**Data availability.** All relevant data are available from the corresponding author upon reasonable request.

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

## Acknowledgements

We thank J. Henley, F. Melchior, G. Bossis, Y. Uchimura, and H. Saitoh for sharing DNA plasmids and H. Leonhardt for the generous gift of mCherry antibodies. We gratefully acknowledge R. Willemsen (Rotterdam, Nl) for the gift of *Fmr1*−/y mice. We also thank L. Davidovic and S. Zongaro for helpful discussion in the initial step of the work and F. Aguila for excellent artwork. We gratefully acknowledge the "Fondation pour la Recherche Médicale" (Equipe labellisée #DEQ20111223747 to S.M.; #DEQ20140329490 to B.B.; #DEQ20110421309 to E.D.), the "Agence Nationale de la Recherche" (ANR-15-CE16-0015-01 to S.M., ANR-12-BSV4-0020, ANR-12-SVSE8-0022, and ANR-15-CE16-0015-02 to B.B., ANR-13-BVS4-0009 to E.D.), the FRAXA foundation to T.M., the French Muscular Dystrophy Association AFM-Téléthon to E.D., the "Jérôme Lejeune" (S.M. and B.B.), and "Bettencourt-Schueller" (S.M.) foundations for financial support. We also thank the French Government for the "Investments for the Future" LabEx "SIGNALIFE" (ANR-11-LABX-0028-01), LabEx "ICST" (ANR-11-LABX-0015-01), the CNRS LIA "Neogenex" and the CG06 (AAP santé), GIS IBiSA (AO 2014) and Région PACA for the Microscopy and Imaging Côte d'Azur (MICA) platform funding. M.P., L.S., and S.C. are fellows from the international PhD "Signalife" LabEx program.

## Author contributions

A.K. performed the majority of fixed granule work and all the neuronal architecture imaging experiments and some biochemistry. C.G. performed most of the molecular cloning, bacterial sumoylation assays, some of the bioinformatic analyses, and some of the endogenous SUMO-protein work. A.K., C.L., F.C., L.S., A.F., M.P., M.Pro., and G.P. prepared neuronal cultures and biochemical tools. M.C. and E.D. performed and analyzed electrophysiology experiments. G.P. prepared viral particles and achieved the FMRP dimerization/sumoylation experiments. T.M. and A.K. performed and analyzed CLIP experiments. S.C. and T.M. performed calcium imaging. F.D.G. and F.B. performed and analyzed smFISH experiments. R.G. performed FMRP structural analysis with help from C.G., F.C., and S.M. performed live-cell imaging experiments. F.Br. provided computational tools to analyze imaging data. D.A. performed and analyzed FLIM experiments. B. B. provided some FMRP coding complementary DNAs and antibodies as well as input for the mRNA work. A.K., C.G., and S.M. contributed to hypothesis development, experimental design, and data interpretation. S.M. provided the overall supervision, the funding, and wrote the article. All authors discussed the data and commented on the manuscript.

## Additional information

**Competing interests:** The authors declare no competing financial interests.

