## [Peer Review File · Nature Communications]

Reviewers' comments:

Reviewer #1 (Remarks to the Author):

This manuscript describes a novel posttranslational regulation of FMRP, the protein that is lost in fragile X syndrome, by Small Ubiquitin-related Modifier-1 (SUMO-1). They perform biochemical experiments to show that SUMO-1 binds to FMRP at two sites in the N-terminus (K88, K130), and one site in the C-terminus (K614). Further experiments showed that mutation of K88,K130 prevented the sumoylation of FMRP. To explore the relevance of sumoylation, the authors overexpress FMRP constructs in cultured neurons from the *Fmr1*^{-/y} and measure dendritic spine shape and density. They find that while expression of FMRP restores dendritic spine maturity and density, the FMRP K88,K130 mutants do not. In further experiments, the authors show that overexpression of the K88,K130 mutant increases the size of FMRP granules and reduces the association of new FMRP within these granules. Application of the mGluR agonist DHPG induces sumoylation and dissociation of FMRP from granules, and this is impaired in the K88,K130 mutant. Based on these findings, the authors conclude that FMRP sumoylation controls association of FMRP with RNA granules, and that this regulates dendritic spine maturation. To directly test this, they express the K88,K130 mutant into WT neurons and show that this alone induces a dendritic spine phenotype similar to the *Fmr1*^{-/y} neurons.

Overall, the main finding of this work is compelling, and would offer a new angle for research into the function of FMRP. The results from experiments in culture are indicative of a functional effect of FMRP sumoylation on dendritic spine morphology and FMRP dimerization. This may well be important for understanding the role of FMRP in brain health, however a much stronger case would be made if the authors could provide any functional evidence in vivo. The experiment looking at FMRP dimerization is a good start, though not sufficient as is.

If the authors could provide evidence that sumoylation affects mRNA binding, dendritic spine morphology, and/or electrophysiological function in the intact brain or brain slice, rather than an overexpression culture system, this would be a good addition to the literature. In addition, there are a few points regarding the experimentation and interpretation of results that should be addressed.

Major comments:

1. In Fig.1 the authors show that FMRP is sumoylated in the brain and cultured neurons by co-immunoprecipitation. This appears to be reduced by the mutation of K130 and K88. Although the examples shown are convincing, it is imperative that these results be quantified across multiple animals and cultures. The authors should provide summary graphs of the FMRP-SUMO interaction from multiple experiments, analyzed with the appropriate statistics. The starting input should be shown on the same blot to compare with the IP fractions.

2. The authors show FMRP sumoylation the brain of very young mice and rats however use mature cultures for the remaining analysis, is FMRP sumoylation developmentally

regulated?

3. For the overexpression experiments, the authors should provide quantitative evidence that they are not expressing more or less FMRP in the mutant cultures (vs WT-FMRP cultures).

4. In the CLIP experiments, the authors need to show whether mRNA association is altered in the K88,K130 mutant (which is used for the rest of the study) and not just the K88,K130,K614 mutant.

5. The experiments tracking the size and kinetics of FMRP particles is interesting, however the assertion that these are "RNA granules" needs more evidence. Is there RNA present in these particles? Are other RNA granule proteins in these particles? How can the authors be sure that the K88,K130 mutant FMRP isn't forming artificial clumps due to overexpression?

6. The experiments showing DHPG affects FMRP kinetics is interesting, and the authors speculate that this is changing mRNA association. This needs to be shown. How does this fit with the data shown in Fig.2 which shows that sumoylation doesn't change mRNA association?

7. Does FMRP sumoylation affect FMRP localization in neurons? Is the number of granules different at synapses when FMRP is sumoylated?

8. The effect of FMRP sumoylation on FMRP homodimerisation shown in Fig.5 is very interesting, but this is only one experiment. This should be replicated and quantified across multiple animals, at an age that is consistent with the culture experiments.

Minor comment:

As the authors describe, several mutations in residues close to the FMRP SUMO active sites have been described in FXS patients. The authors speculate that these could interfere with FMRP sumoylation and this could contribute to FXS pathophysiology. What would be the effect of re-introducing these FMRP mutants (A145S and F126S) in Fmr1 KO neurons? Would these mutants affect FMRP sumoylation in response to mGluR5 stimulation or dendritic spine morphology?

Reviewer #2 (Remarks to the Author):

The authors present evidence from in vivo and in vitro assays that the FMRP protein is modified by SUMO at three lysine residues (K88, 130 and 614) in neurons. Through complementation studies in FMR1-/- neurons, they provide evidence that sumoylation of FMRP is required for its function in regulating dendritic spine maturation and elimination. Through live cell imaging approaches and in vitro biochemical assays, they demonstrate that sumoylation of FMRP affects its homodimerization and the stability of its association

with RNA granules. They also demonstrate that FMRP sumoylation is enhanced by activation of mGlu5 receptors. Collectively, their studies reveal that activity-dependent regulation of FMRP sumoylation plays a key role in neuronal maturation and function.

Overall, this is a well written manuscript with complementary experimental findings that support a new and exciting role for sumoylation in regulating FMRP function and neuronal cell maturation. The findings have important implications because of the role of FMRP in fragile X-syndrome and also because of the emerging, but still poorly understood, roles for SUMO in neurons. There are a number of minor but important issues that when addressed, will improve the strength of the findings and overall conclusions:

- 1) In Figure 1b, c and d, it would be helpful to include western blots demonstrating the IPs in fact precipitated the expected antigens (1b: anti-FMRP; 1c and d: anti-SUMO).
- 2) From the analysis presented in Figure 1h, it is questionable that K88 is actually a SUMO conjugation site. Based on the anti-FMRP blot, levels of SUMO-modified FMRP are not obviously reduced in the single K88R mutant, and they are not further reduced in the K88/130R double mutant compared to the K130R single mutant. The authors should at least comment on this, and also on whether or not they have any functional analysis with the K130R single mutant to support the importance of K88.
- 3) The authors need to include western blot data demonstrating the relative expression levels of the FMRP WT and K/R mutants used in the analysis shown in Figures 2a-d. Without this information, it is not clear that each of the individual proteins was actually expressed, or if differences in expression levels may contribute to the observed findings.
- 4) Western blots revealing relative expression levels should also be included with the analysis in Figure 3a.
- 5) In Figures 4e-g, there needs to be a positive control demonstrating activation of the mGlu5R receptor.
- 6) An anti-SUMO western blot demonstrating equally efficient SUMO pulldowns is also needed in Figure 4e (tubulin is not a useful control).
- 7) Western blots are also needed to demonstrate relative expression levels of FMRP wild type and K/R mutants in experiments shown in Figure 6. Expression levels relative to endogenous FMRP would also be informative.

Responses to reviewers

Reviewer #1

We thank the referee for the insightful comments and constructive suggestions. S/he states, “*the main finding of this work is compelling, and would offer a new angle for research into the function of FMRP.*” and that “*This may well be important for understanding the role of FMRP in brain health*”. However, his/her main concern is that “*a much stronger case would be made if the authors could provide any functional evidence in vivo*” regarding the role of FMRP sumoylation.

To provide additional evidence for a functional role of FMRP and be consistent with the rest of the study, we performed electrophysiological assays on *Fmr1*-KO neurons expressing the WT or the non-sumoylatable form of GFP-FMRP and showed that the mutant GFP-FMRP is not only altering spine density and maturation but also basal synaptic transmission. This is now included in the manuscript as the **Supplementary figure 3**. These new sets of results together with the other additional and initial data included in the revised manuscript provide a clear demonstration that this activity-dependent post-translational mechanism represents meaningful advances to:

- Understand the functional consequences of FMRP sumoylation in a physiological context;
- Better dissect the molecular regulation of FMRP-mRNA trafficking;
- Get additional insights into the emerging activity-dependent roles of neuronal sumoylation.

The referee has also raised a few points regarding the experimentation and interpretation of the results. We carried out several experiments and addressed most of these points strengthening the initial message of the study. The reviewer’s questions/comments are in italics.

1. In Fig.1 the authors show that FMRP is sumoylated in the brain and cultured neurons by co-immunoprecipitation. This appears to be reduced by the mutation of K130 and K88. Although the examples shown are convincing, it is imperative that these results be quantified across multiple animals and cultures. The authors should provide summary graphs of the FMRP-SUMO interaction from multiple experiments, analyzed with the appropriate statistics. The starting input should be shown on the same blot to compare with the IP fractions.

Following the Referee’s suggestion, we have now included a quantitative analysis of the sumoylated form of FMRP over the total level of FMRP in rat and mouse brains as well as in mouse cultured neurons (**Supplementary figure 1c**). The data show that the ratio sumoylated FMRP on the total amount of FMRP is similar between the three conditions tested with a mean ratio of those measurements of $4.19 \pm 0.43\%$.

We have also added two additional experiments showing FMRP or SUMO1 immunoprecipitates probes for FMRP with their respective starting input allowing the direct comparison with the IP fractions (**Supplementary figure 1d,e**). We have also included additional Input/IP controls for SUMO1 to demonstrate the potent effect of the desumoylation blocker NEM (**Supplementary figure 1f**).

2) The authors show FMRP sumoylation the brain of very young mice and rats however use mature cultures for the remaining analysis, is FMRP sumoylation developmentally regulated?

We thank the reviewer for raising this interesting point about the developmental regulation of FMRP sumoylation. We decided to use brain of young rodents as well as mature neuronal cultures obtained from embryonic pups to assess the sumoylation of FMRP since they both present a good ratio between SUMO-FMRP and FMRP levels which can be appreciated in the **Reviewer figure**

1a,b. Indeed, FMRP is regulated during neuronal maturation with the sumoylated form of FMRP being clearly visible at all the maturation stages and at stages classically used to assess dendritic spine morphology (DIV 15-18).

We also showed that the expression of FMRP is developmentally regulated (**Reviewer figure 1b**) in good agreement with previously published work (*Bonaccorso CM et al, 2015, Int. J. Devl Neuroscience*) demonstrating that FMRP expression in the mouse brain is maximum for several days after birth followed by a decreased expression level in the adult brain. In addition, we reported in the past that the sumoylation level is also high after birth (**Reviewer figure 1c**; *Loriol et al, PLoS ONE, 2012*). Therefore, we used early post-natal time points when both FMRP and SUMO1 reach their highest protein expression levels to assess the sumoylation of FMRP in rodent brains. The time frame used here is also consistent with key developmental stages of synapse formation and elimination reported in rodents (*Semple, B et al, 2013, Prog. Neurobiol.*).

Reviewer fig. 1: Developmental regulation of FMRP and sumoylation. (a,b) Representative immunoblots anti-FMRP (Ab#056) of neuronal (a) or brain (b) extracts at different developmental stages prepared in the presence of the cysteine protease inhibitor NEM to prevent desumoylation. (c) Developmental profile of SUMO1-modified substrates from the embryonic day E9 to the post-natal day P14 and the adult (Ad) stage. β -actin loading control is also shown. Adapted from our previous work (*Loriol et al, PLoS ONE 2012*).

3) For the overexpression experiments, the authors should provide quantitative evidence that they are not expressing more or less FMRP in the mutant cultures (vs WT-FMRP cultures).

We have now included immunoblots showing the expression levels of GFP-FMRP in the different conditions of transduction used in the study. This is now included in the **Supplementary figure 2** and **Figure 7**.

4) *In the CLIP experiments, the authors need to show whether mRNA association is altered in the K88,K130 mutant (which is used for the rest of the study) and not just the K88,K130,K614 mutant.*

The referee suggests that we perform additional CLIP experiments with a form of FMRP bearing only the two N-terminal K88 and K130 mutations. We initially performed CLIP experiments using the full non-sumoylatable form of FMRP (*that includes the N-terminal K88R and K130R mutations*) and showed that there was no difference with the WT indicating that these three K-to-R mutations do not alter the ability of FMRP to associate with the assessed mRNAs. Therefore, we feel that assessing mRNA association with the mutant that only includes the K88R and K130R mutations are unlikely to yield meaningful data.

5) *The experiments tracking the size and kinetics of FMRP particles is interesting, however the assertion that these are “RNA granules” needs more evidence. Is there RNA present in these particles? Are other RNA granule proteins in these particles? How can the authors be sure that the K88,K130 mutant FMRP isn't forming artificial clumps due to overexpression?*

This is a critical point and we thank the reviewer to bring this potential pitfall forward. We have now extensively investigated the dendritic granules as suggested by the reviewer. We first performed smFISH experiments with Fluorescent Stellaris probes specific for the exogenous GFP mRNA and for the previously reported endogenous mRNA targets of FMRP, PSD-95 and CaMKII. These data are now included as the **Figure 3a-c** of the revised manuscript and show that the dendritic granules containing either the WT or the mutated form of GFP-FMRP are also labelled by the three separate Stellaris probes indicating that these granules can indeed transport mRNAs.

We further characterized these granules and performed colocalisation assays using antibodies specific for four different proteins (S6, FXR1, Staufen 1 and 2) that are described to be RNA granule protein components (*Kanai et al 2004 Neuron 43:513-25; Elvira et al, 2006 Mol Cell Proteomics. 5:635-51*). The imaging data indicate that all four proteins show some colocalization with both WT and mutated GFP-FMRP-positive granules. These data are now included in the new **Figure 3d-g** of the revised manuscript.

Together with the initial CLIP experiments, these new sets of data unambiguously confirm that the granules containing either the WT or the mutated form of GFP-FMRP are not artificial clumps but dendritic mRNA granules.

6) *The experiments showing DHPG affects FMRP kinetics is interesting, and the authors speculate that this is changing mRNA association. This needs to be shown. How does this fit with the data shown in Fig.2 which shows that sumoylation doesn't change mRNA association?*

We agree with the referee that the fact that a DHPG application speeds up the dissociation of FMRP from RNA granules is interesting and suggest that the mGlu5R-dependent release of FMRP from the RNA granules could precede the mRNA dissociation. Furthermore, we do not think these results are to be in opposition to the data from Figure 2. Indeed, the CLIP data revealed that the set of mRNAs bound to the WT or the non-sumoylatable form of FMRP are similar which correlates well with the new FISH data. The CLIP experiments do not demonstrate that sumoylation changes mRNA association to FMRP but rather that these specific lysine mutations do not preclude the binding of mRNAs to the mutated protein. We have now further discussed this point in the revised manuscript.

7) *Does FMRP sumoylation affect FMRP localization in neurons? Is the number of granules*

different at synapses when FMRP is sumoylated?

The Reviewer underlines the importance of a deeper analysis of the impact of FMRP sumoylation on FMRP localization. We did not identify any difference in the subcellular localization of the WT and mutated form of GFP-FMRP upon expression in *Fmr1*-KO neurons. We showed that FMRP sumoylation triggers the dissociation of FMRP from the dendritic granules only for the WT form of the protein (**Figure 5**). However, we cannot think of any experimental way of tracking the sumoylated form of FMRP since we cannot visualize, nor distinguish SUMO-FMRP from its unmodified form in fixed or living neurons.

Regarding the number of FMRP-containing granules, we measured their frequency along WT and K88,130R GFP-FMRP-expressing *Fmr1*-KO dendrites (**Reviewer figure 2**). Interestingly, the density of dendritic mRNA granules are comparable for the WT and mutated form of GFP-FMRP (**Reviewer figure 2**), while the surface of the granules expressing the K88,130R mutant was significantly increased for WT GFP-FMRP expressing granules (**Fig. 3h,i**). The combination of these data indicates that the activity-dependent regulation of the size of mutant GFP-FMRP containing granules is lost and that the sumoylation of FMRP is essential to control its dissociation from dendritic mRNA granules.

Reviewer figure 2: Preventing FMRP sumoylation does not change the frequency of FMRP-positive dendritic mRNA granules. Histogram shows the average frequency of WT and K88,130R GFP-FMRP granules in *Fmr1*-KO dendrites after 48h and 72h of expression.

8) *The effect of FMRP sumoylation on FMRP homodimerisation shown in Fig.5 is very interesting, but this is only one experiment. This should be replicated and quantified across multiple animals, at an age that is consistent with the culture experiments.*

We agree with the referee that the experiments showing the effect of FMRP sumoylation on its homodimerisation are extremely interesting. These dimer sumoylation/dissociation assays are however purely *in vitro* experiments with a reconstitution of purified FMRP homodimers prior to their *in vitro* sumoylation to assess whether FMRP sumoylation can act as a trigger for the dissociation of these dimers. We have performed these *in vitro* assays several times with consistent results showing a clear involvement of FMRP sumoylation as a dissociation trigger. However, we do not see any alternatives to perform such experiments either in cultured neurons or *in vivo*.

Minor comments:

As the authors describe, several mutations in residues close to the FMRP SUMO active sites have been described in FXS patients. The authors speculate that these could interfere with FMRP sumoylation and this could contribute to FXS pathophysiology. What would be the effect of re-

introducing these FMRP mutants in Fmr1 KO neurons? Would these mutants affect FMRP sumoylation in response to mGluR5 stimulation or dendritic spine morphology?

We agree with the reviewer that the future experiments we suggested in the discussion section are of interest and could explain how some missense FXS mutations could impair the mGlu5R-dependent regulation of FMRP sumoylation. We can, for instance, predict that in the near future, these experiments (*measuring granules size and assessing the morphology and density of dendritic spines in Fmr1-KO neurons*) could be seen as a functional test to validate the pathological involvement of potential point mutations identified within the N-terminal region of FMRP from patients with a FXS phenotype. However, we do not think that these experiments will change the overall conclusions of the present study.

Reviewer #2

We are grateful to this referee for his/her enthusiasm towards our work. S/he states “*Overall, this is a well written manuscript with complementary experimental findings that support a new and exciting role for sumoylation in regulating FMRP function and neuronal cell maturation.*” and that “*The findings have important implications because of the role of FMRP in fragile X-syndrome and also because of the emerging, but still poorly understood, roles for SUMO in neurons.*” However, the referee has “*a number of minor but important issues that when addressed, will improve the strength of the findings and overall conclusions*”. The reviewer’s questions/comments are in italics.

1) In Figure 1b, c and d, it would be helpful to include western blots demonstrating the IPs in fact precipitated the expected antigens (1b: anti-FMRP; 1c and d: anti-SUMO).

We do agree with the referee’s comments and have now included an additional set of immunoprecipitation/immunoblots controls. These data are now included in the revised manuscript as the **Supplementary figure 1 d-f**.

2) From the analysis presented in Figure 1h, it is questionable that K88 is actually a SUMO conjugation site. Based on the anti-FMRP blot, levels of SUMO-modified FMRP are not obviously reduced in the single K88R mutant, and they are not further reduced in the K88/130R double mutant compared to the K130R single mutant. The authors should at least comment on this, and also on whether or not they have any functional analysis with the K130R single mutant to support the importance of K88.

The referee is right in mentioning that it is not obvious that the K88 residue is a SUMO site in the bacterial SUMO assays from the **figure 1**. To better understand the functional relevance of these lysine residues, we have now included additional functional assays on *Fmr1*-KO and WT neurons using single K88R or K130R mutated GFP-FMRP constructs (**Supplementary figure 2 and Figure 7**). These additional data revealed that the single K88 mutant behaves similarly to the WT form of GFP-FMRP whereas the integrity of the K130 residue is critical to the regulation of spine density but not for the maturation of dendritic spines *per se*. These data therefore indicate that the sumoylation of both lysine residues is somehow essential for FMRP function on spine density and maturation. We have now included these data in the revised manuscript.

3) The authors need to include western blot data demonstrating the relative expression levels of the FMRP WT and K/R mutants used in the analysis shown in Figures 2a-d. Without this information, it is not clear that each of the individual proteins was actually expressed, or if

differences in expression levels may contribute to the observed findings.

We have now included immunoblots showing the relative expression levels of the constructs used in the different conditions of transduction used in the study. This is now included in the **Supplementary figure 2** and **Figure 7**.

4) Western blots revealing relative expression levels should also be included with the analysis in Figure 3a.

We have to apologize here since we cannot provide such immunoblots. Indeed, we used a transfection method to express the GFP-FMRP constructs and the maximum number of neurons transfected is not sufficient to give a detectable signal in Western blot. However, the imaging sessions were all achieved using the same zoom, the same laser power / PMT settings across the different conditions tested to be as consistent as possible and limit the variability inherent to this kind of experiments.

5) In Figures 4e-g, there needs to be a positive control demonstrating activation of the mGlu5R receptor.

To answer this point, we imaged the effect of a DHPG activation on the variation of the intracellular concentration of Calcium ions in living WT and *Fmr1*-KO neurons using the ratiometric Fura-2 probe. These data are now included in the **Supplementary figure 5** of the revised manuscript.

6) An anti-SUMO western blot demonstrating equally efficient SUMO pulldowns is also needed in Figure 4e (tubulin is not a useful control).

We do thank the reviewer for raising this point, which allowed us to identify a mistake in the labelling of the **Figure 4e** as the IP was performed using anti-FMRP antibodies and probed for SUMO1 and not the opposite. The control for the FMRP IP is now included in the figure (**Figure 5e** of the revised manuscript) alongside the inputs for both FMRP and Tubulin.

7) Western blots are also needed to demonstrate relative expression levels of FMRP wild type and K/R mutants in experiments shown in Figure 6. Expression levels relative to endogenous FMRP would also be informative.

We have now included immunoblots showing the relative expression levels of the constructs. This is now included in the **Figure 7**.

Concluding statement: We do believe that the sets of data provided here *i.e.*, *the mGlu5R-dependent sumoylation of FMRP as a central regulator of FMRP interactions within dendritic mRNA granules to participate in the activity-dependent control of dendritic spine elimination and maturation*, will greatly contribute to the fields of Neurosciences and Cell biology. We think that this work will generate intense interest from many labs worldwide that aim at further understanding the functional regulation of FMRP but also by providing additional resources to neuroscientists to better assess the physiological relevance of the sumoylation process in neurons. Finally, we thank again the reviewers for their constructive suggestions that helped us to greatly improve the quality of the manuscript.

REVIEWERS' COMMENTS:

Reviewer #1 (Remarks to the Author):

The authors should be commended for the additional experiments performed to strengthen their in vitro data. The added immunoblotting is good additional support for the initial finding that FMRP is SUMOylated. However, this reviewer still has concerns about the lack of evidence of a functional role of FMRP SUMOylation in vivo. In addition, while it is commendable that the authors added electrophysiological data, it is worrisome that these data appear to conflict with the core hypothesis of the study. Indeed, the authors assert that the expression of the SUMOylation mutant FMRP results in Fmr1 KO phenotype of increased dendritic spine density and decreased spine maturity, while the mEPSC data show increased mEPSC amplitude and decreased frequency (the opposite of the published Fmr1 KO phenotype). That this discrepancy is not acknowledged or discussed casts doubts on the interpretation of the spine morphology results.

In general, the current evidence from overexpression of SUMOylation mutant FMRP constructs in neuron culture is not strong enough to overcome the lack of investigation of function in an intact brain. Were there to be a more extended investigation of the in vivo relevance of FMRP SUMOylation it would greatly strengthen the manuscript.

Reviewer #2 (Remarks to the Author):

The authors have done a thorough and excellent job of addressing concerns and comments raised during the first review. They have added important controls, and new experimental data on developmental regulation of FMRP sumoylation. All together, the work provides valuable new insight into the regulation of FMRP and its role in maturation of dendritic spines, as well as new insights into activity-dependent sumoylation in neurons.

Responses to Reviewers: *The reviewers' comments are in italics.*

Reviewer #1

The authors should be commended for the additional experiments performed to strengthen their in vitro data. The added immunoblotting is good additional support for the initial finding that FMRP is SUMOylated. However, this reviewer still has concerns about the lack of evidence of a functional role of FMRP SUMOylation in vivo. In addition, while it is commendable that the authors added electrophysiological data, it is worrisome that these data appear to conflict with the core hypothesis of the study. Indeed, the authors assert that the expression of the SUMOylation mutant FMRP results in Fmr1 KO phenotype of increased dendritic spine density and decreased spine maturity, while the mEPSC data show increased mEPSC amplitude and decreased frequency (the opposite of the published Fmr1 KO phenotype). That this discrepancy is not acknowledged or discussed casts doubts on the interpretation of the spine morphology results.

In general, the current evidence from overexpression of SUMOylation mutant FMRP constructs in neuron culture is not strong enough to overcome the lack of investigation of function in an intact brain. Were there to be a more extended investigation of the in vivo relevance of FMRP SUMOylation it would greatly strengthen the manuscript.

We thank the **Reviewer 1** for acknowledging that the set of data we provided during the revision is a 'good additional support for the initial finding that FMRP is SUMOylated.' However, S/he has concerns about the electrophysiological data included in the **Supplementary figure 3** of the revised manuscript.

Our data only reveal that FMRP expression in *Fmr1*-KO neurons modifies basal synaptic transmission probably via both pre- and post-synaptic modifications. To our knowledge, there are no available data on mEPSCs recorded from FMRP **WT**-expressing *Fmr1*^{-y} cultured hippocampal neurons. However, to answer the reviewer's point on this particular point, we have further discussed the results in the revised manuscript (**p9**) and added three additional references reporting that data comparing mEPSC properties in WT and *Fmr1*^{-y} brain slices show either a decrease, an increase or no changes in their amplitudes or frequencies, depending on the brain area recorded, the age of the animals and/or the associated genetic background. Finally, we do agree with this reviewer that future experiments aiming at investigating the physiological consequences of FMRP sumoylation *in vivo* will be of interest. One way of achieving such experiments would be to design a Knock-in mouse line in which the FMRP-SUMO-target lysines would be mutated into arginine residues to allow for the expression of a non-sumoylatable version of FMRP *in vivo*. While we clearly appreciate the potential add-on of such mouse line to assess the physiological consequences of FMRP sumoylation, we do not think it will change the overall conclusion of the current study.

Reviewer #2

The authors have done a thorough and excellent job of addressing concerns and comments raised during the first review. They have added important controls, and new experimental data on developmental regulation of FMRP sumoylation. All together, the work provides valuable new insight into the regulation of FMRP and its role in maturation of dendritic spines, as well as new insights into activity-dependent sumoylation in neurons.

We are grateful to this referee for his/her strong support to our work.